# Rag GTPases and phosphatidylinositol 3-phosphate mediate recruitment of the AP-5/SPG11/SPG15 complex

Jennifer Hirst[1], Geoffrey G. Hesketh[2], Anne-Claude Gingras[2,3], and Margaret S. Robinson[1]

**Adaptor protein complex 5 (AP-5) and its partners, SPG11 and SPG15, are recruited onto late endosomes and lysosomes. Here we show that recruitment of AP-5/SPG11/SPG15 is enhanced in starved cells and occurs by coincidence detection, requiring both phosphatidylinositol 3-phosphate (PI3P) and Rag GTPases. PI3P binding is via the SPG15 FYVE domain, which, on its own, localizes to early endosomes. GDP-locked RagC promotes recruitment of AP-5/SPG11/SPG15, while GTP-locked RagA prevents its recruitment. Our results uncover an interplay between AP-5/SPG11/SPG15 and the mTORC1 pathway and help to explain the phenotype of AP-5/SPG11/SPG15 deficiency in patients, including the defect in autophagic lysosome reformation.**

## Introduction

Adaptor protein (AP) complexes are a family of five heterotetramers that select cargo for transport from one membrane compartment of the cell to another. AP-1 and AP-2, the most extensively studied of the five complexes, facilitate the formation of clathrin-coated vesicles, with AP-1 acting at the trans-Golgi network (TGN) and early/recycling endosomes and AP-2 acting at the plasma membrane. Although the other AP complexes are (or can be) clathrin independent, many of the interactions that were first reported for AP-1 and AP-2 have been found to apply to other APs as well. For instance, recruitment of AP-1 onto membranes requires the small GTPase ADP-ribosylation factor 1 (ARF1), and ARF1 also contributes to the recruitment of AP-3 onto early/recycling endosomes and AP-4 onto the TGN. Recruitment of AP-2 appears to be ARF independent but requires a phosphoinositide, phosphatidylinositol 4,5-bisphosphate. The recruitment of all four complexes is thought to be by coincidence detection, with several moderate- to low-affinity interactions ensuring they bind to the right membrane. In addition, APs 1, 2, 3, and 4 all bind to similar types of sorting signals on cargo proteins through conserved interactions involving their medium and small subunits (Dell'Angelica and Bonifacino, 2019; Robinson, 2015; Sanger et al., 2019).

The fifth AP complex, AP-5, differs from the other AP complexes in several respects. First, although, like all APs, it consists of two large subunits, a medium subunit, and a small subunit, the sequences of these subunits are sufficiently divergent to be mostly undetectable as homologues using sequence-based tools such as BLAST (Hirst et al., 2011). Even highly conserved regions of the subunits, such as the domains that bind to sorting signals, are different in AP-5, indicating that cargo recognition (assuming it occurs) must be by a different molecular mechanism. Another difference is that, unlike the other AP complexes that localize to intracellular membranes (1, 3, and 4), AP-5 does not require ARF1 for its recruitment (Hirst et al., 2011). In addition, whereas the other complexes are heterotetramers that only transiently associate with other proteins, AP-5 is stably associated with two additional proteins, SPG11 (spatacsin) and SPG15 (spastizin; the gene is also called ZFYVE26; Fig. 1 A; Hirst et al., 2011; Słabicki et al., 2010). SPG15 has a FYVE (Fab1/YOTB/Vac1/early endosome antigen 1 [EEA1]) domain, a zinc finger domain that normally binds to phosphatidylinositol 3-phosphate (PI3P), suggesting that, like AP-2, AP-5 recruitment may require a specific phosphoinositide. Indeed, treatment of cells with the phosphatidylinositol 3-kinase (PI3K) inhibitor wortmannin abolishes recruitment of the AP-5/SPG11/SPG15 complex (Hirst et al., 2013). However, most of the PI3P in the cell is associated with early endosomes, while the AP-5/SPG11/SPG15 complex has been localized to late endosomes and lysosomes, although there are reports that it can be found in other locations as well.

Insights into the function of the AP-5/SPG11/SPG15 complex have come from studies on cells and organisms that are missing particular subunits. Knocking down AP-5 subunits in HeLa cells causes the cells to form aberrant endosomes and lysosomes and impairs their ability to retrieve several membrane proteins from

[1]Cambridge Institute for Medical Research, University of Cambridge, Cambridge, UK; [2]Lunenfeld-Tanenbaum Research Institute, Sinai Health System, Toronto, Ontario, Canada; [3]Department of Molecular Genetics, University of Toronto, Toronto, Ontario, Canada.

Correspondence to Margaret S. Robinson: msr12@cam.ac.uk; Jennifer Hirst: jh228@cam.ac.uk.

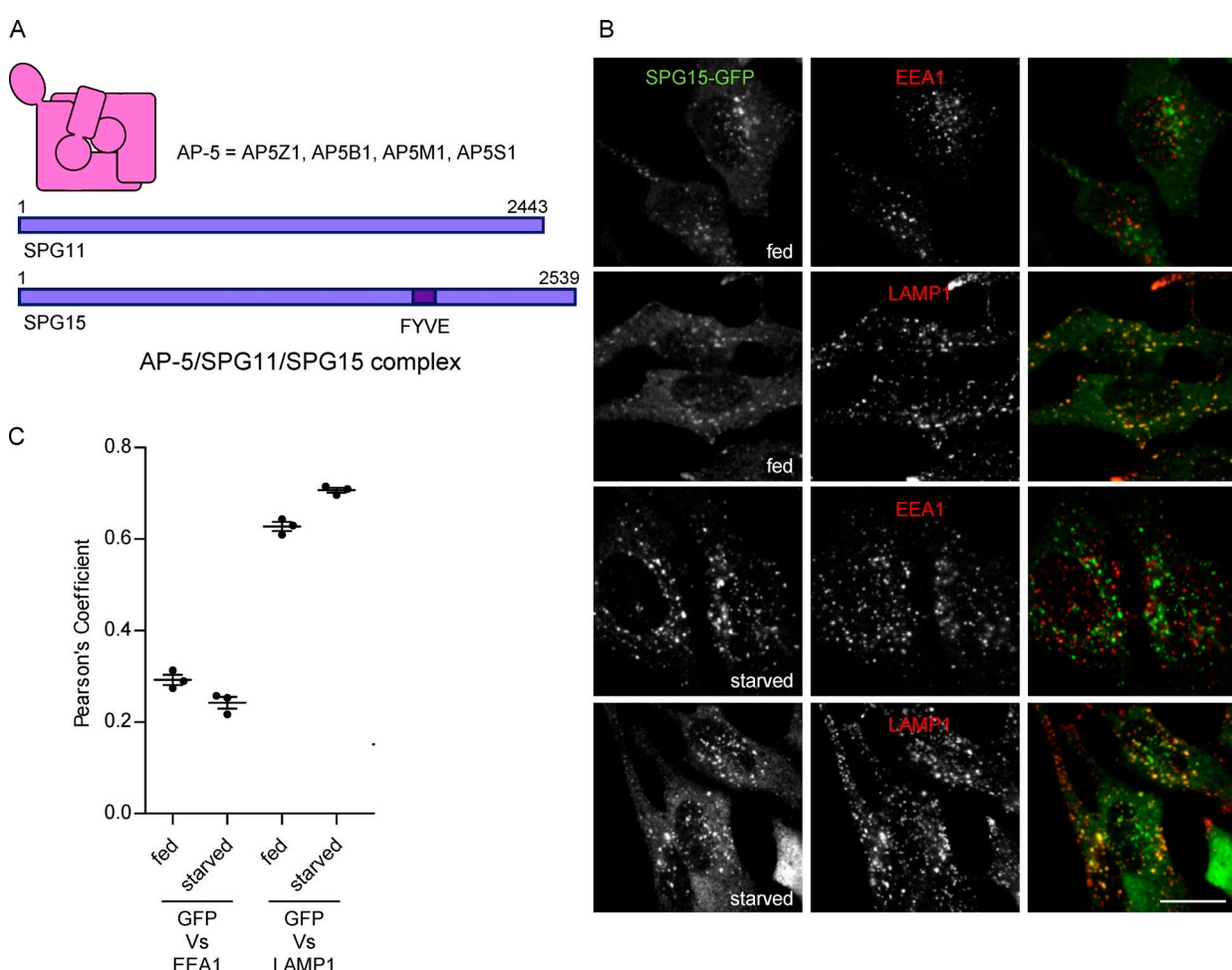

Figure 1.   **Starvation-enhanced recruitment of AP-5/SPG11/SPG15 to late endosomes/lysosomes. (A)** Schematic diagram of the components of the AP-5/SPG11/SPG15 complex. The structure of AP-5 is assumed to be similar to the structures of the other AP complexes, and the complex has a combined molecular weight of ~260 kD. The structures of SPG11 and SPG15 are unknown, but each of these proteins is larger than the entire AP-5 complex, with molecular weights of ~280 kD and ~285 kD, respectively. **(B)** IF labeling of a cell line stably expressing SPG15-GFP (originally created by bacterial artificial chromosome TransgeneOmics; Słabicki et al., 2010). The GFP signal was amplified with anti-GFP, and cells were double labeled for either EEA1 as a marker for early endosomes or LAMP1 as a marker for late endosomes/lysosomes, under either fed or starved conditions (1 h). In this figure and all subsequent figures, starvation was performed in PBS (+Mg+Ca) unless otherwise indicated. Scale bar: 20 μm. **(C)** Quantification of the colocalization between SPG15-GFP and either EEA1 or LAMP1, using Pearson's correlation coefficient. The experiment was performed in biological triplicate (mean indicated; n = 3), and 20 cells were quantified per condition. Error bars represent SEM.

endosomes back to the TGN (Hirst et al., 2011; Hirst et al., 2013; Hirst et al., 2015; Hirst et al., 2018). Aberrant endosomes and lysosomes have also been reported in a number of knockout mouse models (Khundadze et al., 2013; Khundadze et al., 2019; Varga et al., 2015). Loss-of-function mutations in SPG11, SPG15, or AP5Z1 in humans cause a neurodegenerative disorder, hereditary spastic paraplegia, characterized by distal degeneration of corticospinal axons (Boukhris et al., 2008; Hirst et al., 2016; Słabicki et al., 2010). Fibroblasts from patients with mutations in any of these genes accumulate enlarged endolysosomes (Hirst et al., 2013; Hirst et al., 2015; Renvoisé et al., 2014). In addition, lack of either SPG11 or SPG15 in HeLa cells or patient fibroblasts and lack of AP5Z1 in mouse fibroblasts result in a defect in the reformation of lysosomes after fusion with autophagosomes (Chang et al., 2014; Khundadze et al., 2019; Varga et al., 2015). All of these observations point toward a role for the AP-5/SPG11/

SPG15 complex in a retrieval/recovery process that helps to maintain lysosomal homeostasis. It is now clear that lysosomes are not only degradative organelles but also signaling platforms with a key role in nutrient sensing orchestrated by a network of interacting proteins, with Rag GTPases playing a central role (Brady et al., 2016; Hesketh et al., 2018; Lawrence and Zoncu, 2019; Saxton and Sabatini, 2017). The localization of the AP-5/SPG11/SPG15 complex to lysosomes and the phenotype of cells that lack components of the complex raise the possibility that AP-5/SPG11/SPG15 might contribute to metabolic regulation as well as to lysosome maintenance.

In the present study, we set out find out how the AP-5/SPG11/SPG15 complex is recruited onto late endosomes and lysosomes. In doing so, we were able to resolve the localization of the complex and to reconcile some of the different phenotypes that have been reported in loss-of-function studies. We also

uncovered an unexpected connection between the complex and the cell's response to starvation, revealing mechanistic insights into how recruitment is regulated and potentially opening up new therapeutic avenues for patients.

## Results

### Starvation enhances recruitment of the complex

Although we and others have reported that the AP-5/SPG11/ SPG15 complex localizes to late endosomes and lysosomes (Chang et al., 2014; Hirst et al., 2011), some studies have reported localization to centrosomes (Sagona et al., 2010), mitochondria (Murmu et al., 2011), early endosomes (Vantaggiato et al., 2019), and/or the ER (Murmu et al., 2011). These discrepancies are likely to be caused by the low abundance of the complex, the absence of suitable antibodies, and difficulties in tagging the proteins endogenously. Our own studies have been performed mainly on a well-characterized cell line that stably expresses SPG15-GFP under the control of its native promoter for expression at near-physiological levels, originally made using a bacterial artificial chromosome (Słabicki et al., 2010). The SPG15-GFP construct assembles into a complex containing SPG11 and the four subunits of AP-5 (AP5Z1, AP5B1, AP5M1, and AP5S1; Fig. 1 A; Hirst et al., 2011; Hirst et al., 2013), indicating that it is functional. However, expression levels of the construct are somewhat variable from cell to cell, in spite of the cells having been recloned and resorted by flow cytometry. In addition, the GFP signal is so low that normally it needs to be amplified using anti-GFP antibodies (Hirst et al., 2013).

While searching for conditions that might improve the fluorescent signal, we discovered that starved cells were consistently brighter than fed cells (Fig. 1 B). This finding allowed us to take a more thorough and systematic approach to localizing the complex. Consistent with our previous studies, we found that SPG15-GFP was more coincident with lysosome-associated membrane protein 1 (LAMP1), a marker for late endosomes and lysosomes, than with EEA1, a marker for early endosomes, but that the differences were more marked in the starved cells (Fig. 1, B and C). We obtained similar results with stably expressed SPG11-GFP under the control of its endogenous promoter and with transiently expressed HA-SPG15 (Fig. S1, A and B). In every case, there was stronger labeling of the GFP signal associated with the LAMP1 compartment in starved cells than in fed cells. Incubating cells with medium lacking serum, amino acids, or both showed that the increased recruitment was triggered by amino acid starvation rather than serum starvation (Fig. S1 C). Most of the subsequent starvation experiments were performed by incubating the cells in PBS (+Mg+Ca).

### PI3P binding by the SPG15 FYVE domain is necessary for complex recruitment

The SPG15 subunit of the complex contains a FYVE domain (see Fig. 1 A), and we have previously shown that recruitment of the complex is abolished when cells are treated with the PI3K inhibitor wortmannin, suggesting that the complex binds to PI3P via its SPG15 subunit (Hirst et al., 2013). However, most of the PI3P in the cell localizes to early endosomes, while the complex

localizes to a later compartment. Another phosphoinositide, phosphatidylinositol 3,5-bisphosphate [PI(3,5)P2], is most abundant on late endosomes and lysosomes (Ho et al., 2012). Because PI(3,5)P2 is generated by PIKfyve-dependent phosphorylation of PI3P, we considered the possibility that the complex might actually bind to PI(3,5)P2, with PI3P acting as a precursor. However, when we treated cells with either wortmannin or the PIKfyve inhibitor YM201636, we saw very different effects. Whereas wortmannin treatment resulted in the redistribution of SPG15-GFP to the cytosol, YM201636 treatment resulted in increased recruitment of SPG15-GFP onto membranes, with a greater than twofold increase in the apparent size of the spots (Fig. 2, A and B). Similar results were obtained when we localized other components of the AP-5/SPG11/SPG15 complex (Fig. S2). Thus, the AP-5/SPG11/SPG15 complex appears to be a genuine PI3P binder, even though it does not colocalize with the majority of the PI3P in the cell.

To investigate whether the SPG15 FYVE domain might bind to a subset of PI3P associated with a late compartment, we expressed this domain on its own as a GFP fusion protein at a similar expression level to that of the full-length construct (Fig. 2 C). The GFP-FYVE(SPG15) construct was recruited onto EEA1-positive early endosomes rather than LAMP1-positive late endosomes and lysosomes (Fig. 2, D and E). Recruitment was abolished when cells were treated with wortmannin to deplete PI3P, but not when cells were treated with YM201636 to deplete PI(3,5)P2 (Fig. 2 E). In addition, mutations in the SPG15 FYVE domain that are predicted to block binding to PI3P (R1836A) or to block zinc binding (C1867S; Chang et al., 2014) completely abolished the membrane association of GFP-FYVE(SPG15) (Fig. 2 F). In the context of the full-length protein, the same R1836A point mutation prevented the recruitment of HA-SPG15 onto membranes, even in starved cells or cells treated with YM201636 (Fig. 2 G). Thus, the FYVE domain of SPG15 is a typical FYVE domain that, on its own, binds to PI3P associated with early endosomes. However, in the context of the whole complex, SPG15 localizes to late endosomes and lysosomes, presumably by associating with a minor pool of PI3P associated with these organelles. Together, these results indicate that PI3P binding is necessary for the recruitment of the complex but is not sufficient for its localization. Hence, we considered the possibility that there is a second binding site that determines where the complex is recruited.

### Possible role for small GTPases

Small GTPases often contribute to the recruitment of trafficking proteins onto membranes: ARF1 facilitates the recruitment of AP-1, AP-3, AP-4, and coat protein complex I (COPI; Donaldson et al., 2005); Sar1 facilitates the recruitment of coat protein complex II (COPII; Barlowe et al., 1994); Ras-related protein Rab-7 (Rab7) facilitates the recruitment of retromer (Seaman et al., 2009; Rojas et al., 2008); ADP-ribosylation factor–like protein 1 (Arl1) facilitates the recruitment of GRIP domain-containing golgins (Lu and Hong, 2003; Panic et al., 2003); and Arl8 facilitates the recruitment of the homotypic fusion and vacuole protein sorting (HOPS) complex (Khatter et al., 2015). Therefore, we investigated the possible role of three small

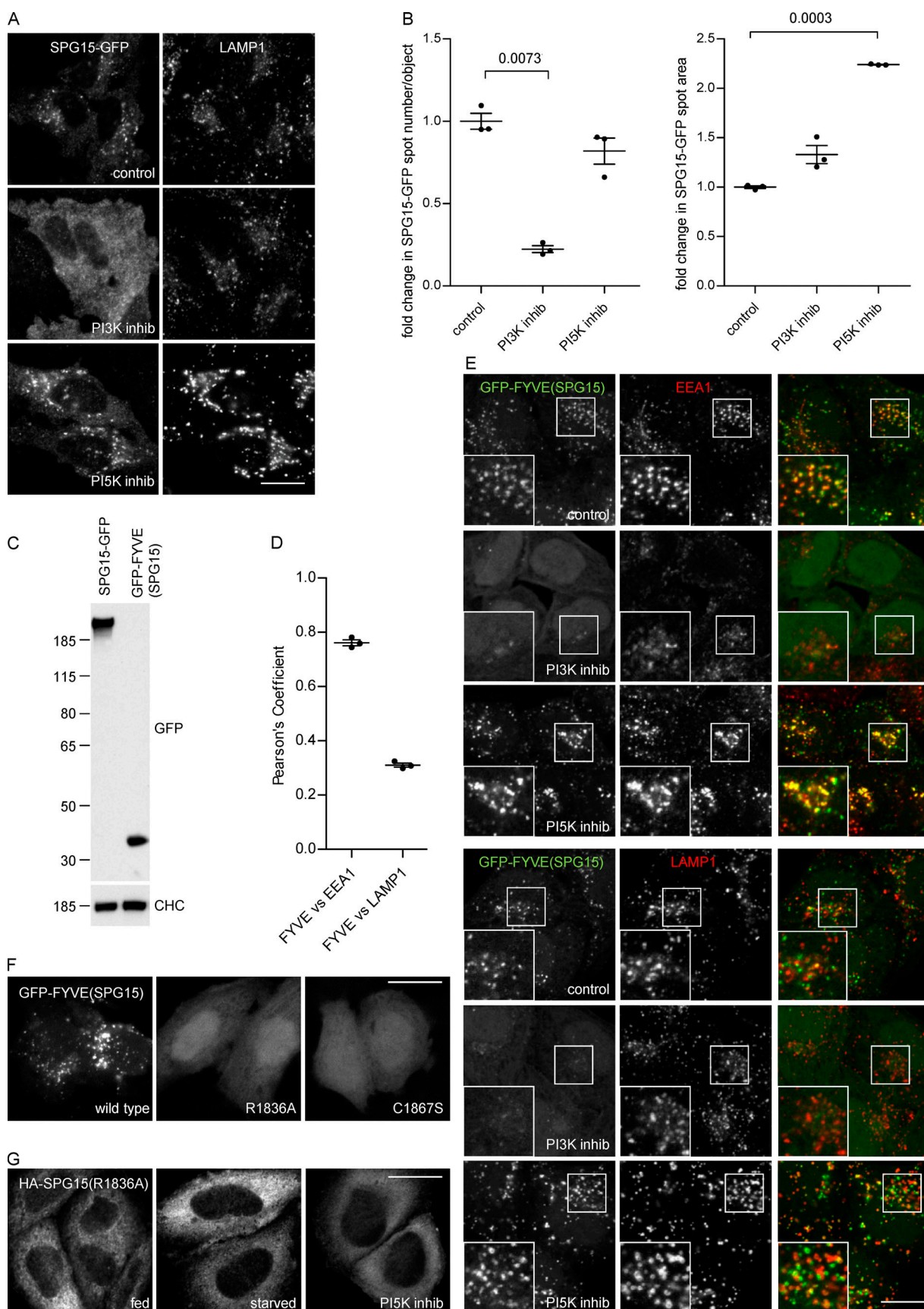

Figure 2. **PI3P binding is necessary for complex recruitment but not sufficient for its localization. (A)** IF double labeling of SPG15-GFP (amplified with anti-GFP) and LAMP1 following treatment with either a PI3K inhibitor (100 nM wortmannin; 1 h) or a PI5K inhibitor (1 μM YM201636; 1 h). Treatment with the

PI3K inhibitor results in a loss of membrane-associated SPG15-GFP, causing the cells to appear brighter because of increased cytosolic labeling, while treatment with the PI5K inhibitor results in enhanced membrane labeling. Scale bar: 20 µm. **(B)** Quantification of the effect of PI3K or PI5K inhibitors on membrane recruitment of SPG15-GFP, showing the fold change of either spot number per object or average spot size relative to control. The slight reduction in the number of spots in the cells treated with the PI5K inhibitor may be due to an indirect effect, such as increased clustering and fusion of endosomes and lysosomes (Bissig et al., 2017). The experiment was performed in biological triplicate (mean indicated; n = 3), and 20 cells were quantified per condition. Error bars represent SEM. **(C)** Western blots of whole-cell lysates from HeLa stable cell lines expressing SPG15-GFP or GFP-FYVE(SPG15), showing similar levels of expression. Clathrin heavy chain (CHC) was used as a loading control. Image is representative of two independent experiments. **(D and E)** IF double labeling and quantification of GFP-tagged SPG15 FYVE domain [GFP-FYVE(SPG15)], stably expressed in HeLa cells, and either EEA1 or LAMP1. Where indicated, cells were treated with either a PI3K inhibitor (100 nM wortmannin; 1 h) or a PI5K inhibitor (1 µM YM201636; 1 h). Scale bar: 20 µm. The quantification of the colocalization between GFP-FYVE(SPG15) and either EEA1 or LAMP1 was performed using Pearson's correlation coefficient. The experiment was performed in biological triplicate (mean indicated; n = 3), and 20 cells were quantified per condition. Note that the isolated FYVE domain of SPG15 predominantly colocalizes with the early endosomal marker EEA1. Error bars represent SEM. **(F)** Localization of GFP-FYVE(SPG15), either wild type, PI3P-binding mutant (R1836A), or zinc-binding mutant (C1867S), transiently expressed in HeLa cells. The inability to bind PI3P or zinc renders the FYVE domain of SPG15 cytosolic. Scale bar: 20 µm. **(G)** Localization of HA-SPG15 with a PI3P-binding mutation (R1836A) transiently expressed in HeLa cells. Where indicated, cells were treated with either a PI3K inhibitor (100 nM wortmannin; 1 h) or a PI5K inhibitor (1 µM YM201636; 1 h). The inability to bind PI3P renders the full-length SPG15 cytosolic. Scale bar: 20 µm.

GTPases that localize to late endosomes and/or lysosomes: Rab7, Rab9, and Arl8. Knocking down these GTPases with siRNA did not produce any obvious effects on the recruitment of SPG15-GFP, nor did manipulating their guanine nucleotide exchange factors (GEFs) or GTPase-activating proteins (GAPs; Fig. 3, A and B; Fig. S3). Thus, none of these GTPases appears to play a critical role in the recruitment of the AP-5/SPG11/SPG15 complex.

Because of our finding that starvation induces the recruitment of the complex to late endosomes/lysosomes, we decided to investigate the role of the Rag GTPases. The Rags are heterodimers consisting of RagA or RagB together with RagC or RagD. Although the Rags have never been implicated in the recruitment of vesicle trafficking proteins, they localize to the right compartment and play a key role in nutrient signaling (Saxton and Sabatini, 2017). We initially focused our attention on RagC because of the availability of a suitable antibody. By immunofluorescence (IF), SPG15-GFP and RagC showed remarkably similar localization patterns (Fig. 3, C and D). Moreover, depletion of RagC by siRNA knockdown resulted in a near-complete loss of SPG15-GFP from the membrane, not only in fed cells but also in starved cells and YM201636-treated cells (Fig. 3, E–G).

### Mammalian target of rapamycin complex 1 (mTORC1) and the AP-5/SPG11/SPG15 complex

The Rag GTPases are well known as essential players in the mTOR signaling pathway. They facilitate the recruitment of mTORC1 onto lysosomal membranes, where the mTOR kinase phosphorylates key substrates (Sabatini, 2017). Thus, loss of RagC would be expected to affect the membrane association of mTORC1 as well as the AP-5/SPG11/SPG15 complex. Moreover, Western blotting showed that knocking down RagC resulted in a loss of RagA, presumably because of destabilization of the heterodimer, and vice versa (Fig. 3 G). Therefore, the situation may be more complicated than a simple binding event, so we performed additional experiments to try to unpick the relationship between the AP-5/SPG11/SPG15 complex and components of the mTORC1 signaling pathway.

The regulation of mTORC1 recruitment involves a number of different proteins, some of which are shown diagrammatically in Fig. 4 A. The Rag GTPase heterodimer is recruited onto

lysosomes by the Ragulator complex, which is anchored to the membrane by N-terminal lipid modifications. The Rags in turn recruit mTORC1 via the mTORC1 regulatory-associated protein of mTOR (RAPTOR) subunit, but only when RagA/B is GTP bound and RagC/D is GDP bound, which occurs under fed conditions (Fig. 4 A; Sabatini, 2017). We tried knocking down other components of the pathway to determine which players are most important for recruitment of the AP-5/SPG11/SPG15 complex. The most dramatic effects were seen when we knocked down either RagC or the Ragulator subunit late endosomal/lysosomal adapter, MAPK and mTOR activator 1 (LAMTOR1; Fig. 4, B and C). Both knockdowns also abolished recruitment of mTOR. We also found that treating cells with two mTOR kinase inhibitors, rapamycin and Torin1, abolished recruitment of SPG15-GFP but not of mTOR (Fig. S4). In contrast, knocking down RAPTOR prevented the recruitment of mTOR but had no effect on the recruitment of SPG15-GFP. Knocking down RagA, the RagC/D GAP folliculin, or the RagA/B GATOR1 GAP subunit nitrogen permease regulator 2–like protein (NPRL2) had more modest effects (Fig. S4).

### Nucleotide loading state determines Rag-dependent recruitment of the complex

The Rag heterodimer has four potential nucleotide loading states: RagA/B$^{GTP}$-RagC/D$^{GDP}$, RagA/B$^{GDP}$-RagC/D$^{GTP}$, RagA/B$^{GDP}$-RagC/D$^{GDP}$, and RagA/B$^{GTP}$-RagC/D$^{GTP}$. Only one of these loading states, RagA/B$^{GTP}$-RagC/D$^{GDP}$, facilitates recruitment of mTORC1, and this is sometimes referred to as the "active state" of the Rag heterodimer, which predominates in fed cells, while the RagA/B$^{GDP}$-RagC/D$^{GTP}$ heterodimer is referred to as the "inactive state" (Brady et al., 2016). Kinetic studies suggest that the state with GTP bound to both subunits (RagA/B$^{GTP}$-RagC/D$^{GTP}$) is unstable, while the state with GDP bound to both subunits (RagA/B$^{GDP}$-RagC/D$^{GDP}$) acts as a transition state between active and inactive (Shen et al., 2017).

To determine which nucleotide loading state facilitates the recruitment of the AP-5/SPG11/SPG15 complex, we transfected cells with RagA, RagB, RagC, or RagD in either "GDP-locked" or "GTP-locked" states and then compared the effects of these constructs on the localization of SPG15-GFP and mTOR. Our prediction was that the GTP-locked forms of RagC or RagD

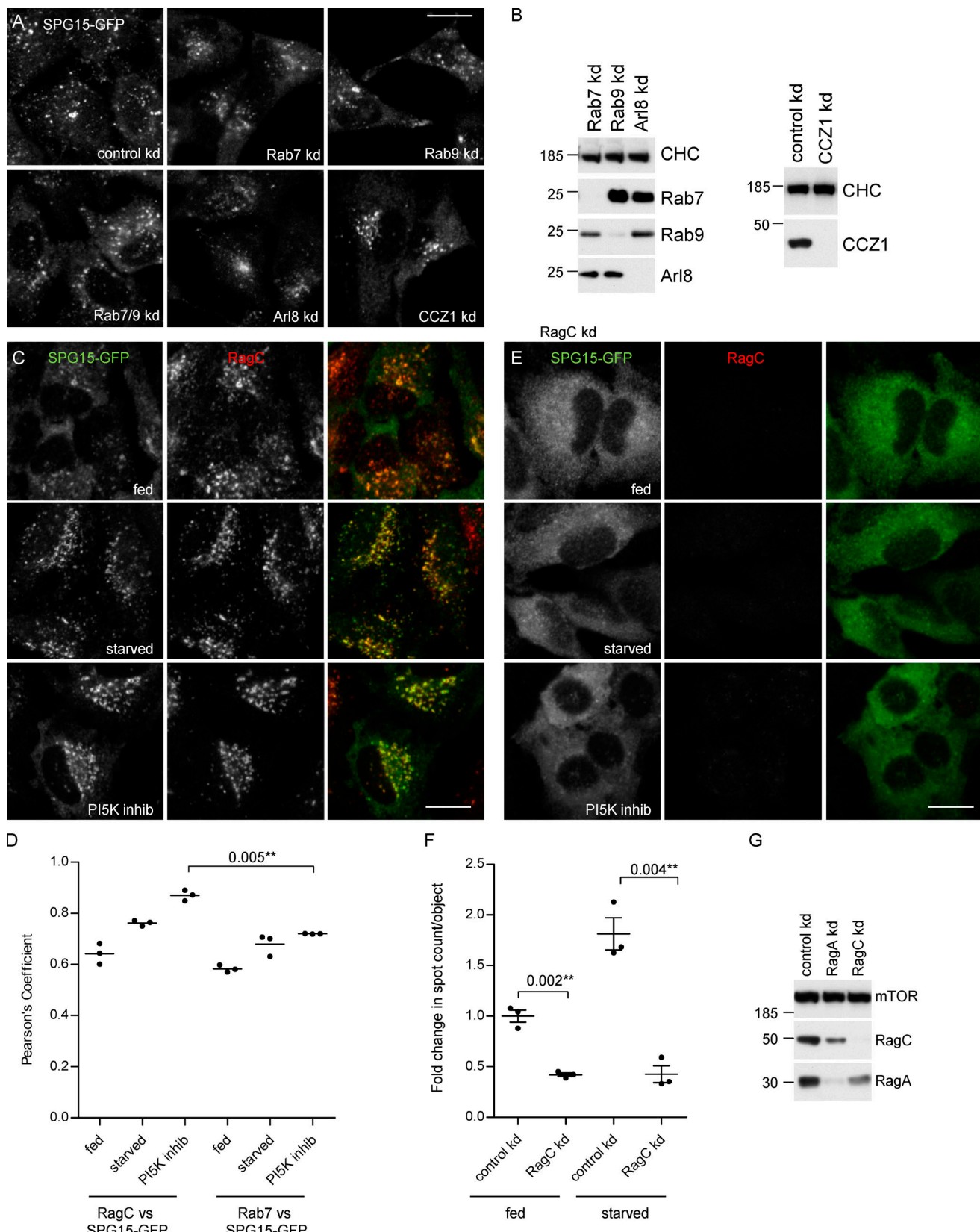

Figure 3. **Possible role for small GTPases. (A)** IF labeling of SPG15-GFP, amplified with anti-GFP, in cells that had been treated with siRNA to knock down Rab7, Rab9, Rab7, and Rab9 together; Arl8; or the Rab7 GEF subunit CCZ1. This was compared with cells transfected with a nontargeting siRNA (control). To better visualize the labeling and any effects of siRNA depletion, all cells were starved for 1 h. None of these knockdowns had a significant effect on the membrane recruitment of SPG15-GFP, although, in some cases, there were effects on its localization (e.g., knocking down Arl8 results in a more perinuclear

distribution, consistent with its known role in lysosome positioning; Pu et al., 2015). Scale bar: 20 µm. **(B)** Western blots of whole-cell lysates from SPG15-GFP cells treated with siRNAs as indicated. Clathrin heavy chain (CHC) was used as a loading control. **(C)** IF double labeling of SPG15-GFP (amplified with anti-GFP) and RagC, either under fed conditions, starved for 1 h, or treated with a PI5K inhibitor (1 µM YM201636; 1 h). Scale bar: 20 µm. **(D)** Quantification of the colocalization between SPG15-GFP and either RagC or Rab7, using Pearson's correlation coefficient. The experiment was performed in biological triplicate (mean indicated; n = 3), and 20 cells were quantified per condition. Error bars represent SEM. **(E)** IF labeling of SPG15-GFP (amplified with anti-GFP) in cells that had been treated with siRNA to knock down RagC. Where indicated, cells were either starved (1 h) or treated with a PI5K inhibitor (1 µM YM201636; 1 h). The RagC knockdown caused the SPG15-GFP to become almost completely cytosolic. Scale bar: 20 µm. **(F)** Quantification of the fold change in spot count per object in SPG15-GFP–expressing cells that were treated with siRNA to knock down RagC or treated with a nontargeting siRNA. Where indicated, cells were incubated under fed or starved conditions (1 h). The experiment was performed in biological triplicate (mean indicated; n = 3), and 20 cells were quantified per condition. Error bars represent SEM. **(G)** Western blots of whole-cell lysates from SPG15-GFP cells treated with siRNAs as indicated; mTOR was used as a loading control. kd, knockdown.

would promote the recruitment of the AP-5/SPG11/SPG15 complex onto membranes, for two reasons. First, this would be consistent with our finding that starvation induces the recruitment of the complex, and second, in other instances where small GTPases recruit trafficking machinery onto membranes, the GTPase needs to be in its GTP-bound state (Wennerberg et al., 2005). Surprisingly, however, we found that it was GDP-locked RagC that most strongly promoted the recruitment of the complex, such that fed cells had just as much membrane-associated SPG15-GFP as starved cells (Fig. 5, A and B, asterisks).

To confirm and extend our microscopy-based findings, we performed proximity-dependent biotinylation and mass spectrometry (BioID; Roux et al., 2012). Cells were generated with stably integrated constructs, allowing inducible expression of the abortive biotin ligase BirA* fused to wild-type, GDP-locked, or GDP-locked RagC. The cells were then incubated with biotin, allowing proteins in close proximity to the BirA*-tagged RagC construct to be covalently biotinylated. Biotinylated proteins were captured by streptavidin affinity purification and identified by mass spectrometry. A total of 235 unique proteins were deemed to be high-confidence proximity interactors (false discovery rate [FDR], ≤1%) across the dataset (see Table S1 for a complete list of protein identifications). Fig. 5 C shows that SPG15 was biotinylated by GDP-locked RagC but not by wild-type or GTP-locked RagC. The mTORC1 RAPTOR subunit was also mainly biotinylated by GDP-locked RagC, consistent with previous studies. Similarly, our IF images show that GDP-locked RagC promoted the recruitment of mTOR, resulting in robust labeling in starved cells as well as fed cells (Fig. 5 A, asterisks). In contrast, the Ragulator LAMTOR1 subunit was biotinylated equally well by wild-type, GDP-locked, and GTP-locked RagC. Although the average spectral counts were higher for RAPTOR and LAMTOR1 than for SPG15, this is likely a reflection of the relative abundance of the three proteins. Proteomic analyses of HeLa cells have shown that RAPTOR and LAMTOR1 are expressed at ∼50-fold and ∼250-fold higher levels, respectively, than SPG15 (Itzhak et al., 2016). None of the other components of the AP-5/SPG11/SPG15 complex were identified, most likely a reflection of their equally low abundance, although it is also possible that SPG15 is closer to RagC than the other five subunits. Interestingly, the PI3K PIK3C3 was also biotinylated by GDP-locked RagC but showed little or no biotinylation in cells expressing GTP-locked wild-type RagC. PIK3C3 is the mammalian homologue of yeast Vps34, and in both yeast and mammalian

cells, this is the kinase that generates PI3P on endosomes (Lindmo and Stenmark, 2006).

From these data, one might predict that GTP-locked RagC would have the opposite effect of GDP-locked RagC. However, our results with this construct were inconclusive, possibly because it was expressed at such low levels (even RagC$^{GDP}$ was expressed at slightly lower levels than endogenous RagC; see Fig. 5 F). Another prediction was that RagD$^{GDP}$ would have a similar effect to RagC$^{GDP}$ because the heterodimer consists of either RagC or RagD together with either RagA or RagB. Interestingly, however, although the relative expression of RagD$^{GDP}$ was somewhat higher than that of RagC$^{GDP}$ and promoted the recruitment of mTOR in starved cells, it did not have any effect on the recruitment of SPG15-GFP (Fig. S5, A–D).

We also observed a very dramatic inhibitory effect of Rag-A$^{GTP}$ on SPG15-GFP, so that it failed to be recruited even in starved cells (Fig. 5, D and E, asterisks). In this case, there was an opposite effect on mTOR, with strong recruitment in starved cells as well as in fed cells, again in line with previous studies showing that GTP-locked RagA prevents mTOR from dissociating from the membrane under starved conditions (Fig. 5, D and E, asterisks; Saxton and Sabatini, 2017). RagA$^{GTP}$ was also noteworthy for being exceptionally highly expressed (Fig. 5 F), resulting in strong cytosolic as well as membrane labeling (Fig. 5 E). However, even when expressed at more moderate levels, RagA$^{GTP}$ had the same effect (Fig. 5 G). RagB$^{GTP}$ also abolished the recruitment of SPG15-GFP (Fig. S5 E). We therefore predicted that RagA$^{GDP}$ or RagB$^{GDP}$ might have the opposite effect and enhance the recruitment of SPG15-GFP. However, neither construct had an appreciable effect on SPG15-GFP recruitment (Fig. S5 E), even though in highly expressing cells RagA$^{GDP}$ abolished mTOR recruitment under fed conditions (Fig. S5 F). The data on nucleotide loading states are summarized in Table 1.

## AP-5/SPG11/SPG15 complex recruitment is dependent on both PI3P and Rags

Overexpression of RagC$^{GDP}$ caused not only SPG15-GFP but also SPG11-GFP and AP5Z1-GFP to be recruited onto membranes, and we have previously shown that depletion of PI3P with wortmannin causes not only SPG15 but also AP-5 to become cytosolic rather than membrane associated (Hirst et al., 2013). All of these results are consistent with the proteins functioning as a stable complex, which requires both PI3P and Rag heterodimers to localize to the correct compartment. Because most of the PI3P is on an early endosomal compartment, we speculated that the

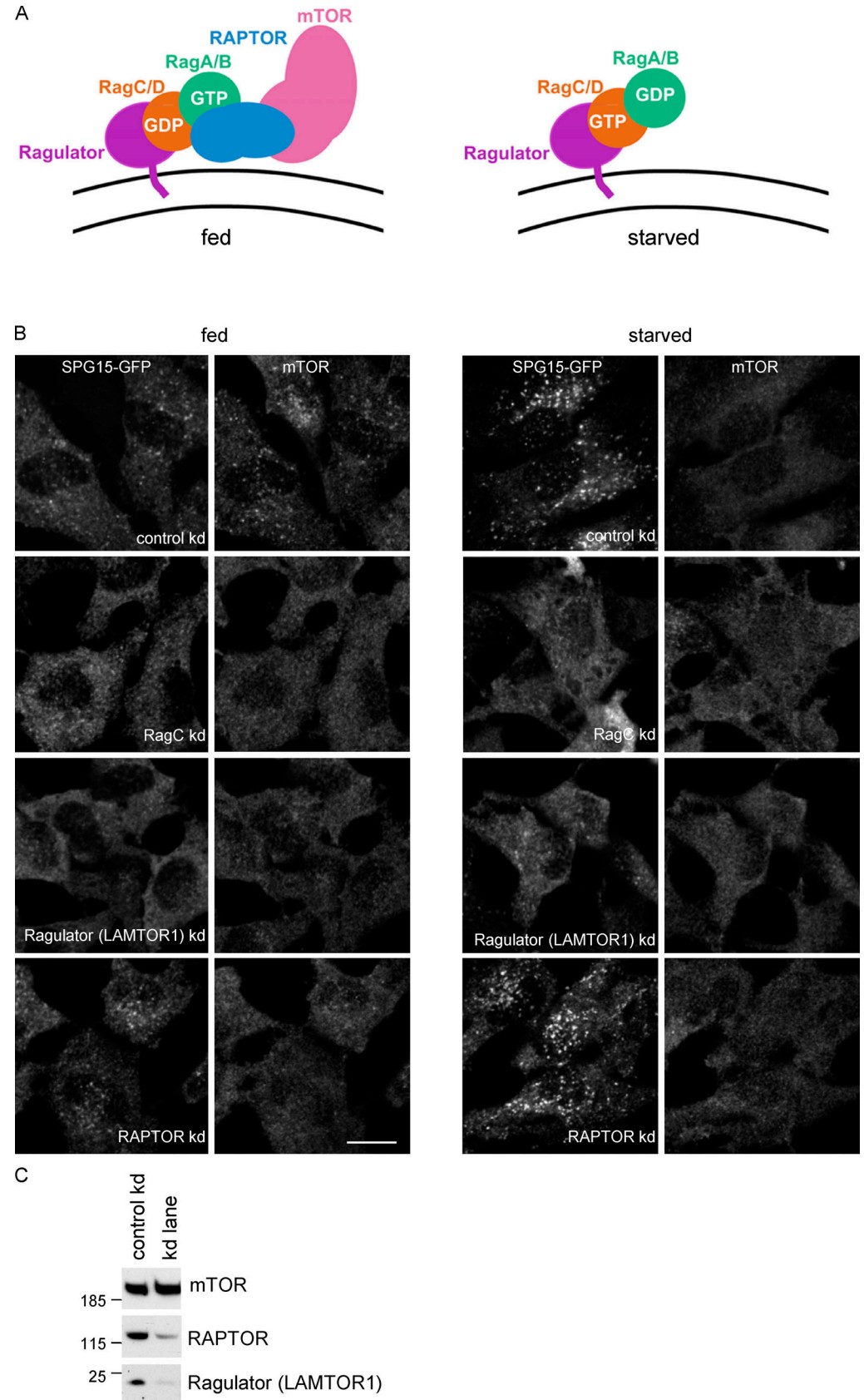

Figure 4. **Membrane recruitment of SPG15 is independent of mTORC1. (A)** Diagram showing the relationships between some of the players in the mTORC1 pathway for nutrient sensing on the lysosome surface. The Ragulator complex recruits the Rag heterodimer, which in turn recruits mTORC1, consisting of RAPTOR, mTOR, and three other subunits; however, Rag binding to mTORC1 only occurs when RagA/B is GTP bound and RagC/D is GDP bound. This

is the nucleotide loading state in fed cells, while starved cells typically have GDP-loaded RagA/B and GTP-loaded RagC/D. **(B)** IF double labeling of SPG15-GFP (amplified with anti-GFP) and mTOR in cells that had been treated with siRNA to knock down either RagC, the Ragulator LAMTOR1 subunit, or RAPTOR, compared with cells transfected with a nontargeting siRNA (control). Knocking down RagC or Ragulator affected the recruitment of both SPG15-GFP and mTOR, while knocking down RAPTOR affected mTOR recruitment only. Scale bar: 20 μm. **(C)** Western blot showing the efficiency of the RAPTOR and Ragulator knockdowns. Shown here is a control versus RAPTOR knockdown, but it is representative of the loading for all knockdowns. kd, knockdown.

complex might initially be recruited onto early endosomes via the SPG15 FYVE domain and then, as the endosome matured, the complex could become stabilized through a Rag-dependent mechanism. However, wortmannin treatment of RagC$^{GDP}$-expressing cells caused SPG15-GFP to become cytosolic, without affecting the localization of the RagC (Fig. 6 A). Moreover, knocking down the Rab7 GEF CCZ1, which blocks early to late endosomal maturation, did not affect SPG15-GFP localization (see Fig. 3 A), while enhancing PI3P levels in cells by PIKfyve inhibition was not sufficient to recruit SPG15-GFP in RagC-depleted cells (see Fig. 3 E). Together, these results suggest that both PI3P and Rag heterodimers need to be present on the same membrane and that the complex is recruited by coincidence detection. This dual dependence on a lipid and a protein may help to explain why it has not been possible to detect Rags in immunoprecipitation or pulldown experiments using components of the AP-5/SPG11/SPG15 complex (Hirst et al., 2013; Słabicki et al., 2010). Similarly, our own more recent attempts to detect AP-5 or SPG15 in Rag pulldowns have been unsuccessful.

The PI3P interaction is a function of the SPG15 FYVE domain, but any of the six subunits of the AP-5/SPG11/SPG15 complex could potentially contribute to the Rag interaction. We therefore investigated the effects of siRNA depletion of either AP-5 or SPG11 on the ability of SPG15-GFP to be recruited onto membranes. Simultaneously knocking down two of the AP-5 subunits (AP5Z1 and AP5M1) led to a slight loss of SPG15-GFP from the cells as a whole (Fig. 6 B) but had little effect on its localization under fed, starved, or RagC$^{GDP}$-expressing conditions (Fig. 6, C and D). Depletion of SPG11 also had only a mild effect on the total amount of SPG15-GFP (Fig. 6 B). However, the SPG11 knockdown caused SPG15-GFP to become much more cytosolic than membrane bound, even in cells that had been starved or transfected with RagC$^{GDP}$ (Fig. 6, C and D). These results suggest that SPG11 contributes to the interaction of the AP-5/SPG11/SPG15 complex with the Rag heterodimer but that AP-5 does not.

Under most conditions, the AP-5/SPG11/SPG15 complex and mTORC1 appear to be in an inverse relationship. Membrane recruitment of the AP-5/SPG11/SPG15 complex is promoted by starvation, while mTORC1 is only recruited in fed cells. Similarly, treatment of cells with Torin1 or rapamycin causes mTORC1 to be locked onto the membrane, while SPG15-GFP is cytosolic (see Fig. S4 C). Intriguingly, however, RagC$^{GDP}$ promotes the recruitment of both. To find out whether AP-5/SPG11/SPG15 and mTORC1 can be recruited onto the same organelles, we performed triple labeling for SPG15-GFP, mTOR, and RagC$^{GDP}$. The three proteins had nearly identical patterns, indicating that a late endosome/lysosome can simultaneously recruit the AP-5/SPG11/SPG15 complex and the mTORC1 complex (Fig. 6 E). However, this does not necessarily mean that they are associated with the same heterodimer. The SPG15-GFP could be

associated with RagA/B$^{GDP}$-RagC$^{GDP}$ heterodimers and mTORC1 with RagA/B$^{GTP}$-RagC$^{GDP}$ heterodimers, which could coexist on the same membrane (Fig. 7; see also Fig. 4 A).

## Discussion

In this study, we show that the recruitment of the AP-5/SPG11/SPG15 complex onto membranes is by coincidence detection, requiring both PI3P and Rag GTPases. When we began the study, we already knew that the FYVE domain of SPG15 was essential for membrane binding (Chang et al., 2014; Sagona et al., 2010). Initially, we speculated that SPG15 might harbor an atypical FYVE domain that could bind to PI3,5P2, which is found on late endosomes and lysosomes, instead of or in addition to PI3P, which is found mainly on early endosomes. However, when we expressed the SPG15 FYVE domain on its own, it localized to early endosomes. In fact, it was unusual in that it was able to be recruited as a single FYVE domain; most other isolated FYVE domain constructs are cytosolic unless they contain two domains in tandem (Hayakawa et al., 2004; Lemmon, 2003). Moreover, treating cells with YM201636, which inhibits PI3,5P2 formation, actually enhanced recruitment of the AP-5/SPG11/SPG15 complex. These findings indicate that the complex binds to the small pool of PI3P associated with late endosomes and lysosomes (Gillooly et al., 2000) and that there is likely to be at least one additional binding partner found on these organelles but not on early endosomes. Therefore, we looked for a possible role for small GTPases, which often facilitate the recruitment of vesicle coat proteins and other trafficking machinery.

When we knocked down various GTPases that localize to late endosomes and lysosomes, the only one with a clear phenotype was RagC. Although the Rag GTPases have not previously been implicated in vesicle trafficking, their role in nutrient sensing fits in well with our finding that recruitment of the complex is enhanced in starved cells. We also found that SPG15-GFP was mainly cytosolic when cells were treated with either of two mTOR inhibitors, rapamycin or Torin1. Both starvation and the mTOR inhibitors induce autophagy, so initially their opposite effects on AP-5/SPG11/SPG15 seemed perplexing. However, rapamycin and Torin1 have been shown to trap mTORC1 on lysosomes (Ohsaki et al., 2010; Settembre et al., 2012), so we propose that the presence of mTORC1 on the lysosome prevents the binding of the AP-5/SPG11/SPG15 complex, either by steric hindrance or through some other mechanism. Similar observations were independently reported by members of the Sabatini laboratory, who performed a proteomic analysis of isolated lysosomes under control conditions, in starved cells, and in cells treated with Torin1. The four AP-5 subunits, SPG11, and SPG15 were among their top hits for proteins whose lysosomal

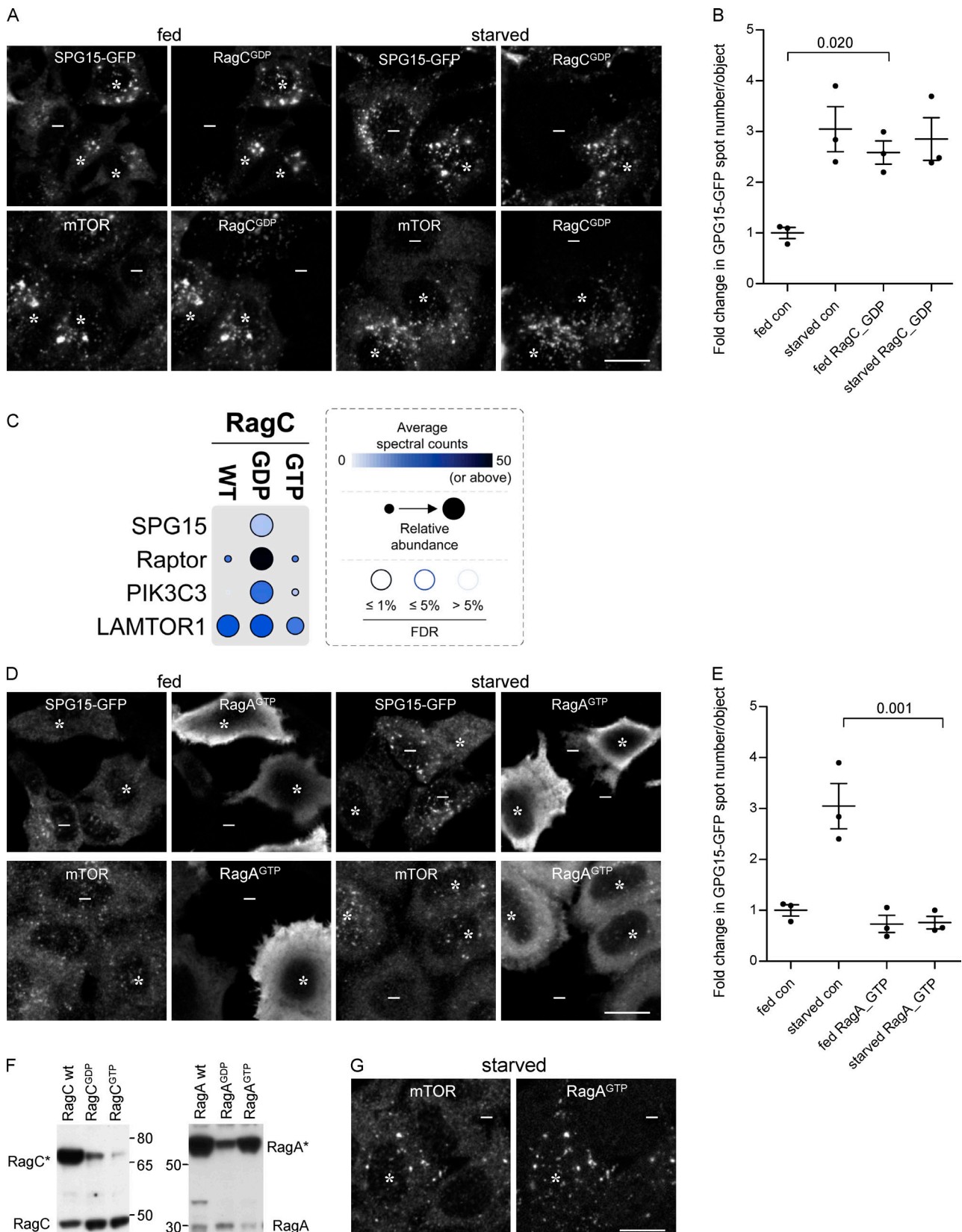

**Figure 5. Recruitment of the complex is dependent on the Rag nucleotide loading status. (A)** IF double labeling of either SPG15-GFP (amplified with anti-GFP) or mTOR in cells that had been transiently transfected with HA-GST–tagged RagC locked in its GDP-bound state. Where indicated, cells were incubated under fed or starved conditions (1 h). The RagC^GDP causes enhanced recruitment of SPG15-GFP onto membranes in fed cells, and it also causes enhanced recruitment of mTOR in starved cells. RagC^GDP-positive cells are marked with asterisks and negative cells with minus signs. Scale bar: 20 μm. **(B)** Quantification

of the fold change in SPG15-GFP spot count per object in RagC$^{GDP}$-expressing cells. The experiment was performed in biological triplicate (mean indicated; $n$ = 3). Error bars represent SEM. **(C)** Dotplot (Knight et al., 2017) of BioID data with C-terminally BirA*-FLAG–tagged RagC (wild-type [WT], GDP-locked, and GTP-locked forms) in HEK293 cells (n.b., only selected proximity interactions are shown; see Table S1 for complete list of protein identifications [in the table, SPG15 is called ZFYVE26]). Dot color represents the abundance (average spectral counts; see inset legend) detected for the indicated prey protein (listed on left) across two biological replicates for the indicated BioID bait (listed on top). Dot outline indicates FDR of interaction as determined by SAINT. Relative abundance detected for given prey proteins across baits is indicated by dot size. **(D)** IF double labeling of either SPG15-GFP (amplified with anti-GFP) or mTOR in cells that had been transiently transfected with HA-GST–tagged RagA locked in its GTP-bound state. Where indicated, cells were incubated under fed or starved conditions (1 h). The RagA$^{GTP}$ prevents recruitment of SPG15-GFP onto membranes in starved cells but causes enhanced recruitment of mTOR in starved cells (RagA$^{GTP}$-positive cells are marked with asterisks and negative cells with minus signs). Scale bar: 20 µm. **(E)** Quantification of the fold change in SPG15-GFP spot count per object in RagA$^{GTP}$-expressing cells. The experiment was performed in biological triplicate (mean indicated; $n$ = 3). Error bars represent SEM. **(F)** Western blots of whole-cell lysates from SPG15-GFP cells transiently transfected with HA-GST–tagged RagC or RagA constructs in either wild-type (wt), GDP-locked, or GTP-locked forms and labeled with antibodies against either RagA or RagC to compare expression levels with each other and with the endogenous proteins. Images are representative of two independent experiments. Note that HA-GST-RagA constructs are generally overexpressed relative to the endogenous protein, and RagC$^{GTP}$ is poorly expressed. **(G)** IF double labeling for mTOR in cells that had been transiently transfected with HA-GST–tagged RagA locked in its GTP-bound state. The cells were imaged 4 d rather than 2 d after transfection, so expression levels were more moderate. The strong effect on mTOR is still apparent. RagA$^{GTP}$-positive cells are marked with asterisks and negative cells with minus signs. Scale bar: 20 µm. con, control.

abundance was increased in starved cells but decreased in Torin1-treated cells (Wyant et al., 2018).

Although the requirement for Rag GTPases could reflect either a direct or an indirect interaction, we suspect that the interaction is direct, in part because of the strong stimulatory effect of GDP-locked RagC on recruitment and the lack of effect of GDP-locked RagD. The sequences of human RagC and RagD are ∼75% identical, differing mainly at their N and C termini. RagC but not RagD is phosphorylated in response to growth factors (Yang et al., 2019), and expression of RagD is more strongly induced by microphthalmia/transcription factor E (MIT/TfE) transcription factors than that of RagC (Di Malta et al., 2017), but no real functional differences between them have been reported, and they recruit mTORC1 equally well. Therefore, we propose that AP-5/SPG11/SPG15 may interact with a binding site that is unique to RagC but that is only available when the RagC is in its GDP-loaded state. This interaction is most likely via SPG11 and/or SPG15, because knocking down SPG11 prevented recruitment of SPG15-GFP, while knocking down AP-5 did not. The strong inhibitory effect of

Table 1. **Effect of GDP-locked and GTP-locked Rags on membrane association of SPG15-GFP and mTOR**

| | SPG15 | | mTOR | |
|---|---|---|---|---|
| Rag | fed | starved | fed | starved |
| Control | + | +++ | +++ | − |
| RagA$^{GDP}$ | nc | nc | ↓$^{hi}$ | nc |
| RagA$^{GTP}$ | ↓↓ | ↓↓ | nc | ↑↑↑ |
| RagB$^{GDP}$ | nc | nc | nc | nc |
| RagB$^{GTP}$ | ↓↓ | ↓↓ | nc | ↑$^{hi}$ |
| RagC$^{GDP}$ | ↑↑↑ | nc | ↑↑ | ↑↑↑ |
| RagC$^{GTP}$ | nc* | nc* | nc* | nc* |
| RagD$^{GDP}$ | nc | nc | ↑ | ↑↑↑ |
| RagD$^{GTP}$ | nc | nc | nc | nc |

+ and −, membrane association in control cells; nc, no change compared to control; ↓↑, change in membrane association; hi, only in high expressers; *, poor expression levels.

GTP-locked RagA and RagB suggests that the RagA/B subunit also needs to be in its GDP-loaded state to bind AP-5/SPG11/SPG15, as shown diagrammatically in Fig. 7. Although the two canonical nucleotide loading states for the Rag heterodimer are RagA/B$^{GTP}$-RagC/D$^{GDP}$, which predominates in fed cells, and RagA/B$^{GDP}$-RagC/D$^{GTP}$, which predominates in starved cells, RagA/B$^{GDP}$-RagC/D$^{GDP}$ also exists, at least as an intermediate (Shen et al., 2017). Precisely how this apparent preference of the AP-5/SPG11/SPG15 complex for both Rags in their GTP-bound state fits in with its increased recruitment during starvation is still unclear. One possibility is that the complex may be recruited at an intermediate stage, before GEF-mediated activation of RagA/B but after GAP-mediated activation of RagC.

What are the functional implications of the Rag interaction, and how might the cell benefit from increased recruitment of AP-5/SPG11/SPG15 during starvation? Several lines of evidence indicate that the AP-5/SPG11/SPG15 complex is involved in the reformation of lysosomes from autolysosomes and endolysosomes (Chang et al., 2014; Khundadze et al., 2019). This pathway would need to be up-regulated in response to starvation, when there is increased lysosomal activity, to ensure that fusion-competent terminal storage lysosomes are constantly being replenished. Indeed, a recent study on the first AP-5–knockout mouse showed that although autophagy was unaffected in the knockouts under basal conditions, it was impaired under stressed conditions, with a defect in autophagic lysosome reformation in starved cells and reduced degradation of an aggregation-prone huntingtin construct in transfected cells (Khundadze et al., 2019).

Our demonstration that the AP-5/SPG11/SPG15 complex is recruited onto the same late endosomal/lysosomal compartment where mTORC1 signaling occurs and that it requires both PI3P and Rag GTPases should settle the uncertainty about where the complex is localized. The only compartment where both PI3P and Rags are present is the late endosome/lysosome, so we suspect that the alternative localization patterns that have been reported are due to antibody cross-reactivity (a particular problem when the antigen is of such low abundance), overexpression artifacts, and/or organelles being in such close proximity that they are difficult to resolve. Our study also adds to the weight of evidence implicating the AP-5/SPG11/SPG15

Figure 6. **SPG15 recruitment is dependent on the simultaneous binding of PI3P and Rags. (A)** IF double labeling for SPG15-GFP (amplified with anti-GFP) in cells that had been transiently transfected with HA-GST–tagged RagC<sup>GDP</sup> (labeled with anti-GST). Where indicated, cells were incubated with the PI3K inhibitor wortmannin (1 h). Note that the inhibitor prevents the RagC<sup>GDP</sup>-dependent recruitment of SPG15-GFP in fed cells. Scale bar: 20 µm. **(B)** Western blots of whole-cell lysates from SPG15-GFP cells treated with siRNA to knock down AP-5 or SPG11, showing knockdown efficiencies. Image is representative of two independent experiments. **(C)** IF double labeling for SPG15-GFP (amplified with anti-GFP) in cells that had been treated with siRNAs against AP-5 or SPG11 and then transiently transfected with HA-GST–tagged RagC<sup>GDP</sup>. The RagC<sup>GDP</sup>-dependent recruitment of SPG15-GFP in fed cells is inhibited by SPG11 depletion,

while the cytosolic labeling of SPG15-GFP is increased, but this does not occur in AP-5–depleted cells. Scale bar: 20 µm. **(D)** Quantification of the fold change in spot count per object in SPG15-GFP cells in cells treated with siRNA to knock down AP-5 or SPG11. Where indicated, cells were incubated under fed or starved conditions (1 h) or transiently transfected with HA-GST-RagC$^{GDP}$. The experiment was performed in biological triplicate (mean indicated; $n$ = 3). Error bars represent SEM. **(E)** IF triple labeling for SPG15-GFP (amplified with anti-GFP) and mTOR in cells that had been transiently transfected with HA-GST–tagged RagC$^{GDP}$ (labeled with anti-HA). Note that the expression of HA-GST-RagC$^{GDP}$ causes the hyperrecruitment of both SPG15-GFP and mTOR onto the same structures. Scale bar: 20 µm. kd, knockdown.

complex in autophagic lysosome reformation, a process that is known to be regulated by mTOR (Yu et al., 2010). Normally, although mTOR signaling is inhibited during starvation, after ∼8 h, the mTOR is reactivated and functional lysosomes are replenished. This process has been shown to be impaired in the absence of the AP-5/SPG11/SPG15 complex, and, intriguingly, a recent study on immortalized embryonic fibroblasts from an AP5Z1-knockout mouse reported less mTOR activity after prolonged starvation (Khundadze et al., 2019). The inverse relationship we describe between the AP-5/SPG11/SPG15 complex and mTORC1 hints at likely feedback mechanisms, although currently it is difficult to explain why the absence of the AP-5/SPG11/SPG15 complex would cause a decrease in mTOR signaling rather than an increase. Thus, there are still many open questions about the interplay between the AP-5/SPG11/SPG15 complex and mTOR signaling. There are also questions about the relationship with PI3P. Players in the autophagy pathway include the PI3K Vps34/PIK3C3 (Kihara et al., 2001), which was one of the hits in our RagC proximity biotinylation assay (see Fig. 5 C). The inositol 5-phosphatase INPP5E, which converts lysosomal PI3,5P2 to PI3P (Hasegawa et al., 2016), has also been implicated in autophagy. Could these enzymes contribute to the increased recruitment of AP-5/SPG11/SPG15 in starved cells?

Perhaps the most fundamental question is, what does the AP-5/SPG11/SPG15 complex actually do? The other AP complexes all select cargo for transport from one membrane compartment to another, so it seems likely that AP-5 and its partners play a similar role. Consistent with this hypothesis, AP-5/SPG11/SPG15-deficient cells show impairment in retrieval of Golgi proteins from late endosomes (Hirst et al., 2018; Khundadze et al., 2019) and in autophagic lysosome reformation (Chang et al., 2014; Khundadze et al., 2019). However, whether these

phenotypes are direct or indirect consequences of the deficiency, whether one is a consequence of the other, and whether the AP-5/SPG11/SPG15 complex actually forms a vesicle coat are still open questions. AP-1, AP-2, and AP-3 have all been localized to budding profiles by immunogold electron microscopy, but AP-5 and its partners are 100–200-fold less abundant than APs 1–3 (Itzhak et al., 2016), so determining whether the complex is associated with endolysosomal buds or tubules is technically challenging. An alternative approach would be to develop an in vitro system to investigate whether binding of AP-5/SPG11/SPG15 to liposomes leads to membrane deformation. Our discovery that recruitment of the complex requires both PI3P and GDP-bound RagC means that such an approach is now potentially feasible. The availability of such a system would also enable us to address other structural questions, such as how the complex interacts with the Rag heterodimer, and whether recruitment onto membranes causes a conformational change, similar to the ones that have been described for AP-1 and AP-2 (Ren et al., 2013; Jackson et al., 2010; Kelly et al., 2014). An allosteric mechanism to block the FYVE domain would help to explain why this domain on its own is able to bind to membranes, while in the context of the whole complex, the FYVE domain is only active if the complex is also able to interact with Rag heterodimers in a particular nucleotide loading state.

The requirement for Rag GTPases places the AP-5/SPG11/SPG15 complex right at the heart of the signaling network that governs the cell's response to starvation, consistent with its role in lysosome maintenance and in autophagic lysosome reformation, processes that are particularly important in neurons (Magalhaes et al., 2016; Varga et al., 2015). Thus, our findings provide new insights into both the function of the AP-5 AP-5/SPG11/SPG15 in normal cells and why its absence leads to spastic paraplegia and could potentially open up new therapeutic approaches, such as the use of autophagy regulators.

## Materials and methods
### Antibodies
The following antibodies were used in this study: rabbit anti-mTOR (2983; Cell Signaling Technology), rabbit anti-RagC (9480; Cell Signaling Technology), rabbit anti-RagA (4357; Cell Signaling Technology), rabbit anti-RAPTOR (2280; Cell Signaling Technology), rabbit anti-LAMTOR1 (8975; Cell Signaling Technology), rabbit anti-folliculin (anti-FLCN; 3697; Cell Signaling Technology), rabbit anti-NPRL2 (37344; Cell Signaling Technology), mouse anti-GFP for IF (ab1218; Abcam), rabbit anti-GFP for Western blotting (gift from Matthew Seaman, University of Cambridge, Cambridge, UK), chicken anti-GFP for triple-labeling IF (ab13970), mouse anti-HA (16B12; Covance),

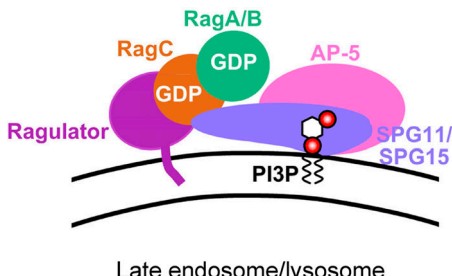

Figure 7. **Speculative model of AP-5/SPG11/SPG15 recruitment.** We propose that the AP-5/SPG11/SPG15 complex binds to two molecules on the late endosomal or lysosomal membrane: PI3P, via the SPG15 FYVE domain, and RagC in its GDP-loaded state, via SPG11 and/or SPG15. The strong inhibitory effect of RagA$^{GTP}$ and RagB$^{GTP}$ suggests that RagA/B also needs to be in its GDP-loaded state.

rabbit anti-GST (in-house), rabbit anticlathrin heavy chain (in-house), mouse anti-AP1G1 (Mab100.3; in-house), mouse anti-EEA1 (610457; BD Transduction Laboratories), rabbit anti-EEA1 (3288; Cell Signaling Technology), rabbit anti-LAMP1 (ab24170; Abcam), rabbit anti-AP5Z1 (HPA035693; Atlas), rabbit anti-SPG15 (FYVE-CENT 8532; Cell Signaling Technology), rabbit anti-Rab9a (gift from Paul Luzio, Cambridge Institute for Medical Research, Cambridge, UK), rabbit anti-Rab7a (EPR7589; Abcam), rabbit anti-Arl8b (ab105694; Abcam), and mouse anti-CCZ1 (sc514290; Santa Cruz Biotechnology). HRP-conjugated secondary antibodies (1:5,000) were purchased from Sigma-Aldrich. Fluorescently labeled secondary antibodies used in this study were Alexa Fluor 488–labeled goat antichicken IgY (A11039), Alexa Fluor 546–labeled donkey antirabbit IgG (A10040), Alexa Fluor 647–labeled donkey antimouse IgG (A31571), Alexa Fluor 488–labeled donkey antimouse IgG (A21202), Alexa Fluor 594–labeled donkey antimouse IgG (A21203), Alexa Fluor 488–labeled donkey antirabbit IgG (A21206), and Alexa Fluor 594–labeled donkey antirabbit IgG (A21207), all purchased from Invitrogen and used at 1:500.

## Constructs
The FYVE domain (residues 1797–1875) of SPG15 was amplified from SPG15 cDNA by PCR, cloned in-frame into Xho1- and BamH1-cut pEGFP_C1 using Gibson Assembly Master Mix (E2611; New England Biolabs), and transferred into pLXINmod using EcoR1 and BamH1 and the Rapid DNA Ligation Kit (11635379001; Roche). pLXINmod is a modified retroviral vector based on pLXIN but with an extended multiple cloning site and was a gift from Andrew Peden (University of Sheffield, Sheffield, UK). The zinc-binding (C1867S) and PI3P-binding (R1836A) SPG15 FYVE domain mutants were constructed using overlapping oligos and Gibson Assembly Master Mix. The HA-SPG15 wild-type and PI3P-binding mutant (R1836A) were a gift from Craig Blackstone (National Institutes of Health, Bethesda, MD; Chang et al., 2014). mCherry-TBC1D5 was a gift from Matthew Seaman (University of Cambridge, Cambridge, UK). pRK5-HA GST Rag constructs were gifts from David Sabatini (Whitehead Institute, Boston, MA). The constructs we used are as follows: pRK5-HA GST RagA T21L (RagA$^{GDP}$; Addgene plasmid 19299), pRK5-HA GST RagA_Q66L (RagA$^{GTP}$; Addgene plasmid 19300), pRK5-HA GST RagB T54L (RagB$^{GDP}$; Addgene plasmid 19302), pRK5-HA GST RagB Q99L (RagB$^{GTP}$; Addgene plasmid 19303), pRK5-HA GST RagC S75L (RagC$^{GDP}$; Addgene plasmid 19305), pRK5-HA GST RagC Q120L (RagC$^{GTP}$; Addgene plasmid 19306), pRK5-HA GST RagD S77L (RagD$^{GDP}$; Addgene plasmid 19308), and pRK5-HA GST RagD Q121L (RagD$^{GTP}$; Addgene plasmid 19309). For all Rag constructs, similar levels of transfection efficiency were achieved, with ~75% of cells expressing 48 h after transfection. However, we observed that there were significantly different levels of expression, with RagA constructs expressed at higher levels than RagC and RagD expressed at higher levels than RagC.

## Cell culture
HeLa M cells and HEK293ET cells were obtained from the European Collection of Authenticated Cell Cultures. HeLa cell lines stably expressing SPG15-GFP, SPG11-GFP, and AP5Z1-GFP were made using bacterial artificial chromosome TransgeneOmics (Słabicki et al., 2010) and extensively characterized by Hirst et al. (2013). In spite of having been recloned and resorted, their expression levels were somewhat heterogeneous. All cells were maintained in DMEM high glucose (D6546; Sigma-Aldrich) supplemented with 10% vol/vol FCS, 4 mM L-glutamine, 100 U/ml penicillin, and 100 µg/ml streptomycin and cultured at 37°C under 5% $CO_2$. Stable cell lines were additionally maintained with 500 µg/ml G418 as appropriate.

Transient DNA transfections were performed using a TransIT-HeLaMONSTER kit (Mirus Bio LLC) according to the manufacturer's instructions. HeLa cells stably expressing GFP-FYVE(SPG15) were created using retrovirus made in HEK293ET cells transfected using TransIT-293 Transfection Reagent (Mirus Bio LLC) according to the manufacturer's instructions. pLXIN plasmids were mixed with the packaging plasmids pMD.GagPol and pMD.VSVG in a ratio of 10:7:3. Viral supernatants were harvested after 48 h, filtered through a 0.45-µm filter, supplemented with 10 µg/ml hexadimethrine bromide (Polybrene; Sigma-Aldrich), and applied directly to the target cells at 37°C. Antibiotic selection for stable expression (500 µg/L G418) was initiated 48 h post-transduction. Clonal cell lines were isolated and selected for expression at near-endogenous levels. Where indicated, cells were treated with 100 µg/ml brefeldin A for 10 min at 37°C or 100 nM wortmannin (PI3K inhibitor), 1 µM YM201636 (PI5K inhibitor), 250 nM Torin1 (4247; Tocris), and rapamycin at 200 ng/ml final concentration in cell culture medium for 1 h at 37°C. Cells were starved by washing three times with PBS(+Mg+Ca). They were then incubated for 1 h in PBS(+Mg+Ca) or in amino acid–free DMEM (048-33575; Wako), with or without 10% (vol/vol) dialyzed FCS (A11-107; PAA) at 37°C. Unless otherwise indicated, the starvation was performed in PBS(+Mg+Ca). The cell lines were routinely tested for the presence of mycoplasma contamination using the MycoAlert Mycoplasma Detection Kit (LT07-318; Lonza) and were also regularly treated with mycoplasma removing agent (093050044; MP Biomedicals), even though they had a negative test result. All chemicals were bought from Sigma-Aldrich unless stated otherwise.

## siRNA-mediated knockdown
Knockdown of AP-5 was achieved by combined siRNA targeting of AP5Z1 and AP5M1 using ON-TARGETplus SMARTpools (AP5Z1, L-025284; AP5M1, L-015523 [C14orf108]; Dharmacon) with a double-hit 96-h protocol. For both hits, the final concentration of siRNA was 30 nM (15 nM AP5M1 + 15 nM AP5Z1), and the second hit was performed 48 h after the first. For all other knockdowns, ON-TARGETplus SMARTpools were used at concentrations of 25–50 nM in a single-hit 3-d protocol. The following siRNAs were used: RPTOR (L-004107), FLJ21439 (SPG11) (L-107138), LAMTOR1 (L-020916), RRAGA (L-016070), NPRL2 (L-015645), FLCN (L-009998), RRAGC (L-017822), RAB9a (L-004177), RAB7a (L-010388), ARL8a (L-016577), ARL8b (L020294), and CCZ1 (L-021482). The reason we targeted Rab7a, Rab9a, and Arl8a + Arl8b is that we know from HeLa cell proteomics that these are the only Rab7, Rab9, or Arl8 paralogues that are detectable (Itzhak et al., 2016). Transfections of siRNA

were performed with Oligofectamine (Thermo Fisher Scientific) according to the manufacturer's instructions and, where indicated, cells were mock treated with a nontargeting siRNA (D-001810; Dharmacon). Three independent experiments were performed, unless otherwise noted in the figure legends. Knockdown efficiencies were determined by Western blotting and showed >90% depletion for most knockdowns and >85% for RAPTOR knockdown (quantified by using ImageJ).

## Fluorescence microscopy

Cells were grown onto 13-mm glass coverslips or MatTek glass-bottom dishes and fixed in 3% formaldehyde in PBS (137 mM NaCl, 2.7 mM KCl, 10 mM $Na_2HPO_4$, and 1.76 mM $KH_2PO_4$, pH 7.4). Formaldehyde-fixed cells were permeabilized with 0.1% saponin and blocked in 0.5% BSA/0.02% saponin in PBS. Primary antibody (diluted in BSA block) was added for 60 min at room temperature. Coverslips were washed three times in BSA block and then fluorochrome-conjugated secondary antibody was added in BSA block for 45 min at room temperature. Coverslips were then washed three times in PBS, followed by a final wash in distilled $H_2O$, before being mounted in ProLong Diamond Antifade Reagent (Thermo Fisher Scientific). For Rab7 labeling, fixed cells were permeabilized with 0.25% Triton X-100 and blocked in 0.5% BSA in PBS. For triple labeling, the following combination was used: rabbit anti-mTOR, mouse anti-HA, chicken anti-GFP, and Alexa Fluor 546–labeled donkey antirabbit IgG, Alexa Fluor 647–labeled donkey antimouse IgG, Alexa Fluor 488–labeled goat antichicken IgY. Widefield and confocal images were captured on a Zeiss LSM 710 confocal microscope on an inverted AxioImagerZ1 using a Zeiss Plan Apochromat 63× oil immersion objective (NA 1.4) and ZEN Black software version 2.3. Images were always processed in the same way with the same exposure times and the same manipulations to optimize brightness and contrast.

Quantification of colocalization, measured using Pearson's correlation coefficient, and of spot number per object or average spot size was performed on subsaturation confocal images (Volocity software 6.3; PerkinElmer). The "Find Spots" algorithm was used to identify and count spots. This is an intensity maxima method combined with a defined radius background subtraction. The automatically calculated detection threshold was manually adjusted using a small subset of images to accurately separate spots from the background. The adjusted threshold was then applied to all of the images during spot counting. Spot size (area) was derived using the Volocity "Find Objects" algorithm. Internally calibrated native format image files were used for accurate measurements. Segmentation of objects from the background was achieved using automatic intensity-based segmentation. The objects that were identified were allocated to individual cells using a cell mask (or manually drawn regions of interest) and were filtered by minimum size to remove single-pixel noise.

At least 20 cells were analyzed for each condition and repeated at least three times. For statistical analysis, data were analyzed by using a two-tailed Student's *t* test. Data distribution was assumed to be normal, but this was not formally tested. Where representative images are shown, the experiment was

repeated at least three times. Statistical analyses of imaging data were performed using GraphPad Prism version 5.01 (GraphPad Software).

## Western blotting

Estimations of protein concentrations were made using a Pierce BCA Protein Assay Kit (Thermo Fisher Scientific). Cells were lysed for Western blot analysis in 2.5% (wt/vol) SDS/50 mM Tris, pH 8. Lysates were passed through a QIAshredder column (Qiagen) to shred DNA and then boiled in NuPAGE LDS Sample Buffer (Thermo Fisher Scientific). Samples were loaded at equal protein amounts for SDS-PAGE, performed on NuPAGE 4–12% Bis-Tris gels in NuPAGE MOPS SDS Running Buffer (Thermo Fisher Scientific). PageRuler Plus Prestained Protein Ladder (Thermo Fisher Scientific) was used to estimate the molecular sizes of bands. Proteins were transferred to nitrocellulose membrane by wet transfer, and membranes were blocked in 5% wt/vol milk in PBS with 0.1% vol/vol Tween-20 (PBS-T). Primary antibodies (diluted in 5% milk) were added for at least 1 h at room temperature, followed by washing in PBS-T, incubation in secondary antibody (also in 5% milk) for 30 min at room temperature, and washing in PBS-T and finally PBS. Chemiluminescence detection of HRP-conjugated secondary antibody was performed using Amersham ECL Prime Western Blotting Detection Reagent (GE Healthcare) and x-ray film. Where representative blots are shown, the experiment was repeated at least three times. Western blots were developed using ECL Prime Western Blotting Detection Reagent (GE Healthcare) and quantified using ImageJ software.

## BioID

BioID was performed essentially as described previously (Hesketh et al., 2017). C-terminally tagged (BirA*-FLAG) RagC constructs were generated by Gateway cloning from a sequence-validated entry vector (NM_022157.4). GTP-locked (Q120L) and GDP-locked (S75N) mutants were generated by PCR mutagenesis and sequence validated. Polyclonal populations of stable HEK293 Flp-In T-REx cells with integrated BirA*-FLAG–tagged constructs were selected and maintained with 200 µg/ml hygromycin B. Parental cell lines were negative for mycoplasma contamination (MycoAlert; Lonza). Cells were grown on 15-cm plates to ~75% confluency, and bait expression and proximity labeling were induced simultaneously by addition of tetracycline (1 µg/ml) and biotin (50 µM) for 24 h. Bait samples (biological duplicates) were compared against 24 independent negative control samples (12 cell lines expressing BirA*-FLAG only and 12 cell lines expressing triple-FLAG only). The specific control samples used in this study were previously published (Chapat et al., 2017). Cells were collected in PBS, and biotinylated proteins were purified by streptavidin-sepharose affinity purification. Proteins were digested on-bead with sequencing-grade trypsin in 50 mM ammonium bicarbonate (pH 8.5). Peptides were then acidified by the addition of formic acid (2% vol/vol final concentration) and dried by vacuum centrifugation. Dried peptides were suspended in 5% (vol/vol) formic acid and analyzed in data-dependent acquisition mode on a TripleTOF 5600 mass spectrometer (SCIEX) inline with a nanoflow electrospray ion

source and nano-HPLC system. Raw data were searched and analyzed within ProHits LIMS (Liu et al., 2010). High-confidence proximity interactions (FDR, ≤1%) were determined through Significance Analysis of INTeractome (SAINT; Teo et al., 2014) implemented within ProHits (see Table S1 for the SAINT output file). Dotplots were prepared in ProHits-viz (Knight et al., 2017). Mass spectrometry data have been deposited as a complete submission to the MassIVE repository (https://massive.ucsd.edu/ProteoSAFe/static/massive.jsp) and assigned the accession number MSV000086151. The ProteomeXchange accession number is PXD021519.

### Online supplemental material

Fig. S1 shows further characterization of the effect of starvation on the AP-5/SPG11/SPG15 complex. Fig. S2 shows the effects of PI3K and PI5K inhibition on different components of the AP-5/SPG11/SPG15 complex. Fig. S3 shows knockdown of small GTPases or their modifying enzymes. Fig. S4 shows effects of different treatments on the AP-5/SPG11/SPG15 complex versus mTORC1. Fig. S5 shows the effect of Rags in different nucleotide loading states. Table S1 lists interactions identified by BioID.

## Acknowledgments

We thank Paul Luzio and members of the Robinson laboratory for reading the manuscript and for helpful discussions.

This work was supported by Wellcome Trust grants 086598 and 214272 to M.S. Robinson and 100140 to the Cambridge Institute for Medical Research; Canadian Cancer Society grants 704301 and 705938 to A.-C. Gingras; Canadian Institutes of Health Research grant FDN 143301 to A.-C. Gingras; and a basic research fellowship from Parkinson Canada to G.G. Hesketh. Mass spectrometry was performed at the Network Biology Collaborative Centre at the Lunenfeld-Tanenbaum Research Institute, a facility supported by Canada Foundation for Innovation funding, by the Government of Ontario, and by Genome Canada and Ontario Genomics (OGI-139).

The authors declare no competing financial interests.

Author contributions: J. Hirst, G.G. Hesketh, and M.S. Robinson conceptualized the experiments; J. Hirst and G.G. Hesketh performed and analyzed the experiments; J. Hirst, G.G. Hesketh, A.-C. Gingras, and M.S. Robinson interpreted the data; and J. Hirst, G.G. Hesketh, and M.S. Robinson wrote the manuscript.

Submitted: 13 February 2020

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

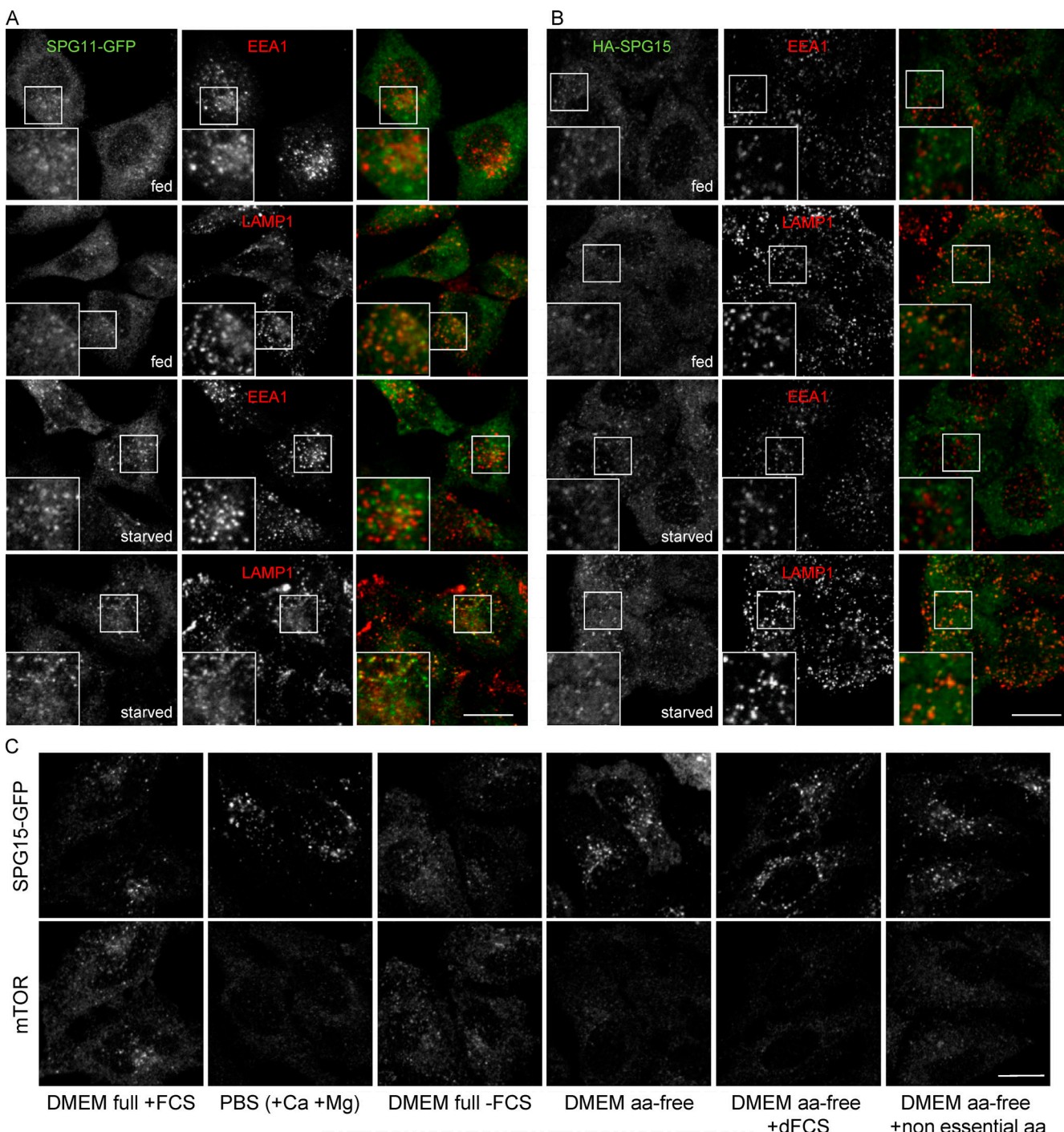

Figure S1. **Further characterization of the effect of starvation on the AP-5/SPG11/SPG15 complex. (A)** IF double labeling of SPG11-GFP (amplified with anti-GFP) and either EEA1 as a marker of early endosomes or LAMP1 as a marker of late endosomes/lysosomes, under either fed or starved conditions (1 h). Scale bar: 20 μm. **(B)** HeLa cells were transiently transfected with HA-SPG15 and double labeled for HA and either EEA1 or LAMP1, under either fed or starved conditions (1 h). Note the starvation-induced enhanced recruitment of both SPG11-GFP and HA-SPG15 to LAMP1-positive late endosomes/lysosomes. Scale bar: 20 μm. **(C)** Localization of SPG15-GFP (amplified with anti-GFP) under control conditions or starved 1 h, using the indicated conditions. dFCS, dialysed FCS. Scale bar: 20 μm.

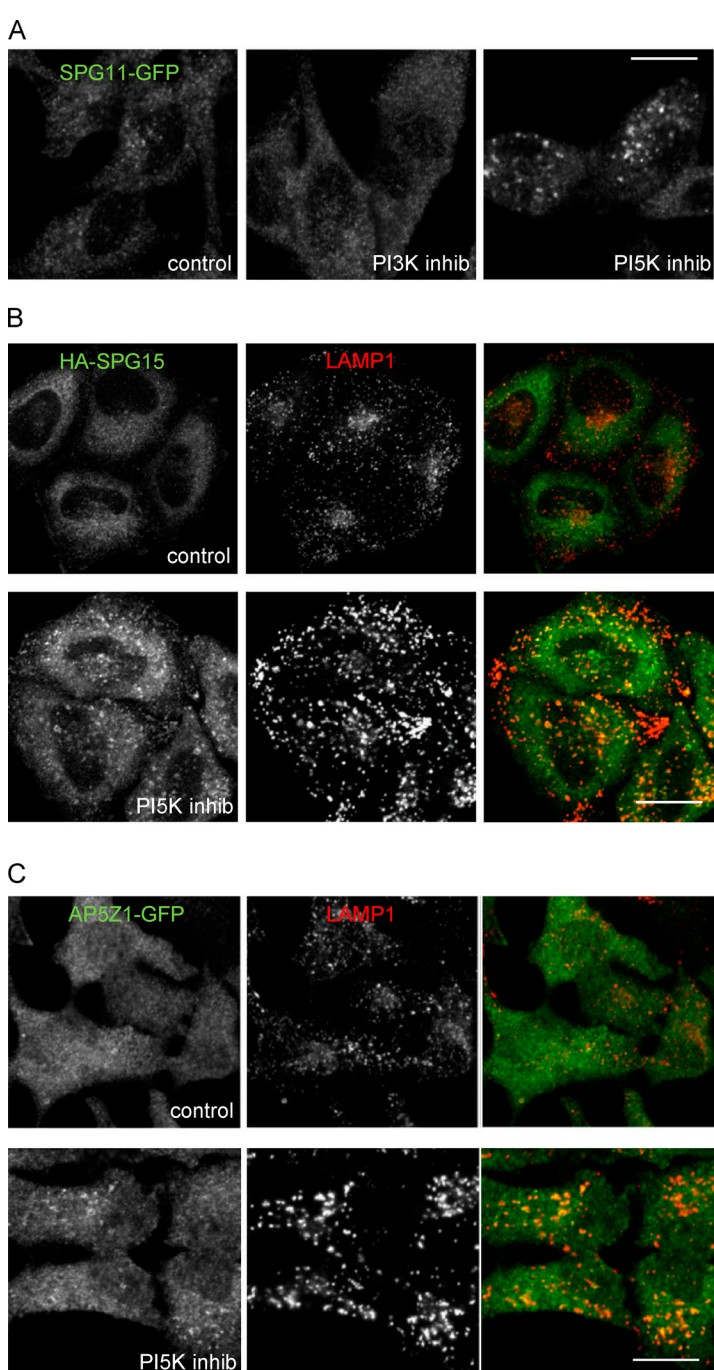

Figure S2. **Effects of PI3K and PI5K inhibition on different components of the AP-5/SPG11/SPG15 complex. (A)** Localization of SPG11-GFP (amplified with anti-GFP) under control conditions or after treatment with either a PI3K inhibitor (100 nM wortmannin; 1 h) or a PI5K inhibitor (1 µM YM201636; 1 h). Like SPG15-GFP, SPG11-GFP is lost from the membrane upon PI3K inhibition, but membrane localization is enhanced upon PI5K inhibition. Scale bar: 20 µm. **(B)** IF double labeling of HA-tagged SPG15 and LAMP, either under control (fed) conditions or after treatment with a PI5K inhibitor (1 µM YM201636; 1 h). Membrane localization of the HA-tagged SPG15 is much more apparent after PI5K inhibition. Scale bar: 20 µm. **(C)** IF double labeling of AP5Z1-GFP (amplified with anti-GFP) and LAMP, either under control (fed) conditions or after treatment with a PI5K inhibitor (1 µM YM201636; 1 h). Membrane localization of AP5Z1-GFP is normally difficult to discern, but it is enhanced by PI5K inhibition. Scale bar: 20 µm.

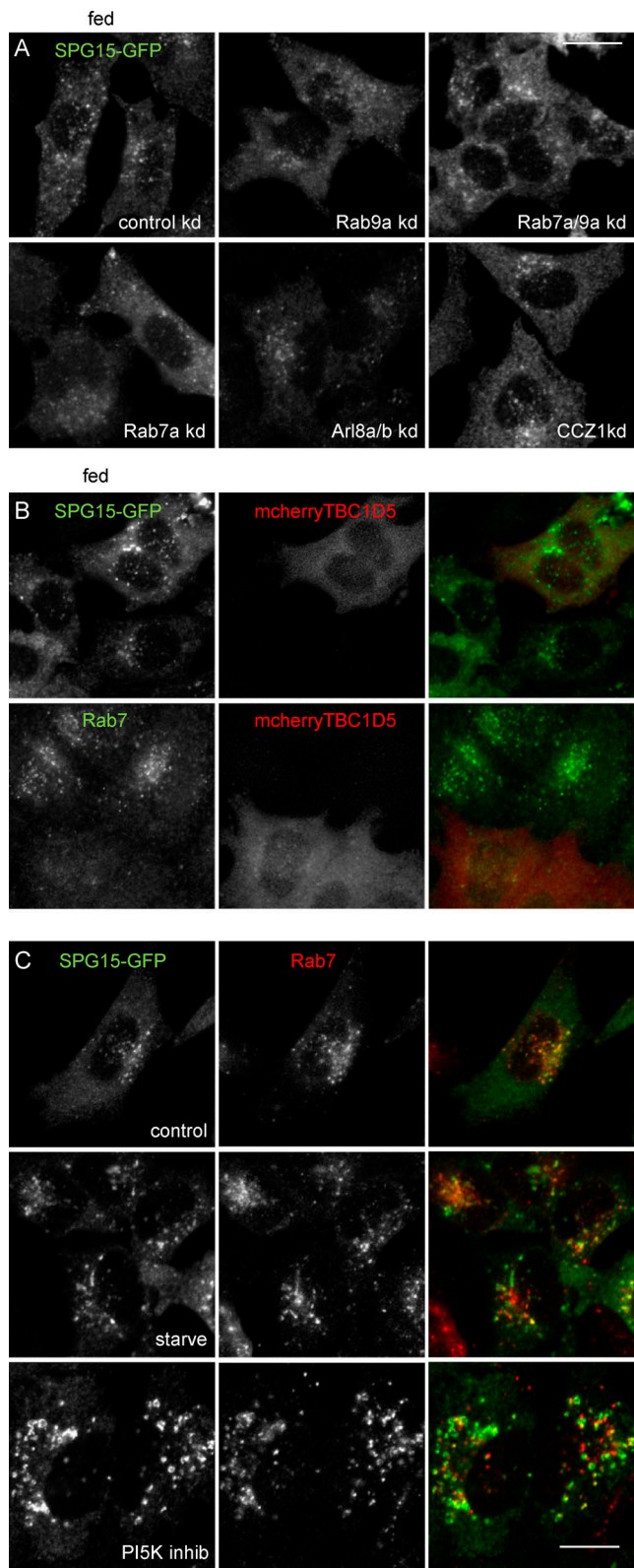

Figure S3. **Knockdown of small GTPases or their modifying enzymes. (A)** IF labeling of SPG15-GFP (amplified with anti-GFP) in cells that had been treated with siRNA to knock down Rab7a, Rab9a, or both together; Arl8a/Arl8b; and CCZ1. This was compared with control cells transfected with a nontargeting siRNA under fed conditions. Fig. 3 shows the same treatments in starved cells. Scale bar: 20 µm. **(B)** IF labeling of SPG15-GFP (amplified with anti-GFP) in cells that had been transfected with TBC1D5, a GAP for Rab7, tagged with mCherry. Note that the overexpression of mcherry-TBC1D5 causes the loss of Rab7 from membranes but has no apparent effect on the localization of SPG15-GFP. Scale bar: 20 µm. **(C)** Localization of SPG15-GFP (amplified with anti-GFP) and Rab7 under control conditions or after treatment with either a PI3K inhibitor (100 nM wortmannin; 1 h) or a PI5K inhibitor (1 µM YM201636; 1 h). Scale bar: 20 µm. kd, knockdown.

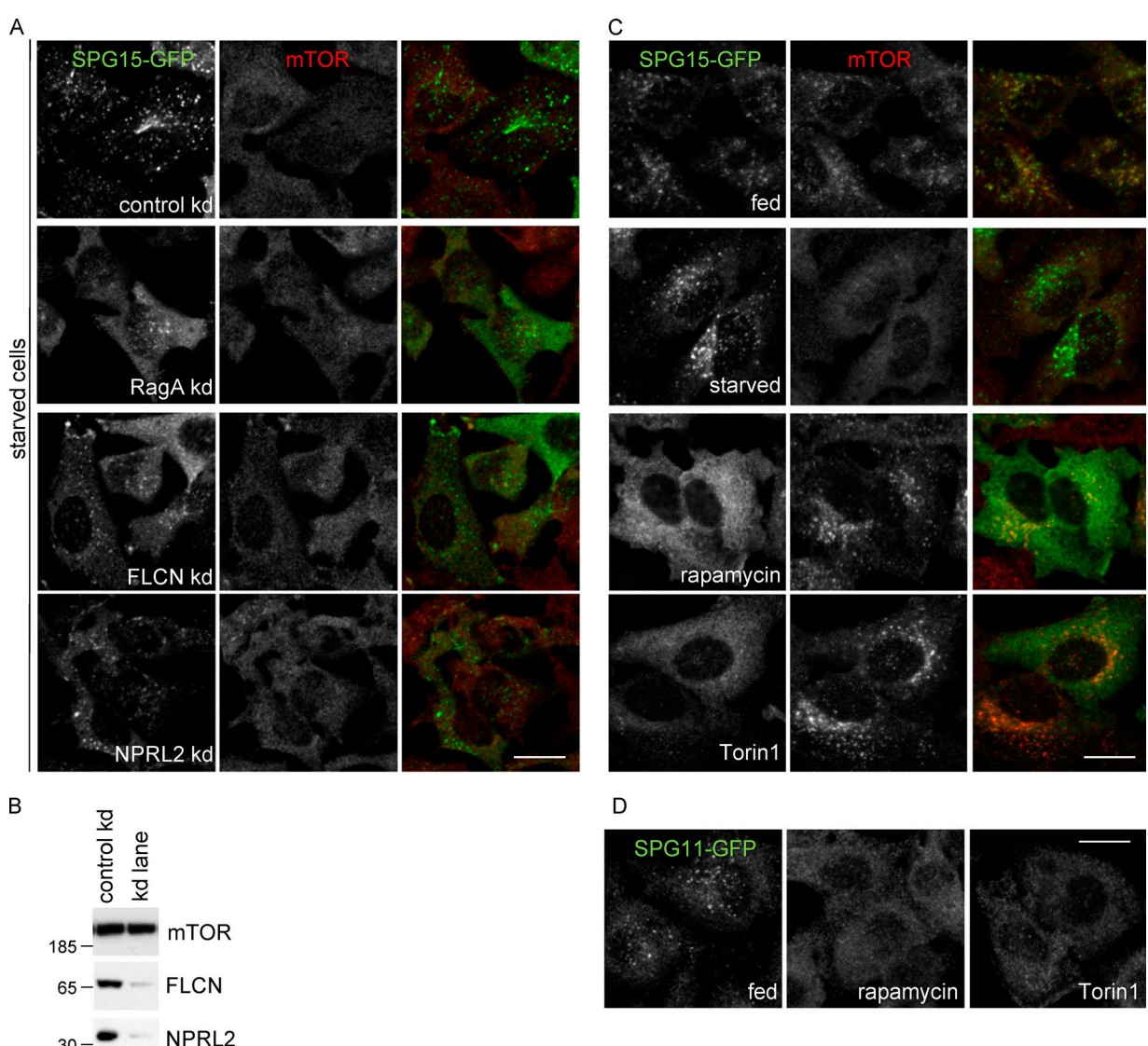

Figure S4. **Effects of different treatments on the AP-5/SPG11/SPG15 complex versus mTORC1. (A)** IF labeling of SPG15-GFP (amplified with anti-GFP) in cells that had been treated with siRNA to knock down RagA, the RagC/D GAP FLCN, or the RagA/B GATOR1 GAP subunit NPRL2. To better visualize the labeling and any effects of siRNA depletion, all cells were starved (1 h). These knockdowns have a slight effect on SPG15-GFP recruitment, but they are not as dramatic as a RagC or Ragulator knockdown. Scale bar: 20 μm. **(B)** Western blots of whole-cell lysates from SPG15-GFP cells treated with siRNA as indicated, with mTOR as a loading control. The blot shows mTOR labeling in a control versus FLCN knockdown, but it is representative of the loading for both knockdowns. The blots are representative of at least two independent experiments. **(C)** IF labeling of SPG15-GFP (amplified with anti-GFP) and mTOR in cells treated in various ways, showing an inverse relationship between the two. Starvation (1 h) increases SPG15 recruitment while preventing mTOR recruitment, while the drugs rapamycin and Torin1 lock mTOR on the membrane and prevent recruitment of SPG15. Scale bar: 20 μm. **(D)** IF labeling of SPG11-GFP (amplified with anti-GFP), showing that, like SPG15-GFP, its membrane association is inhibited by rapamycin and Torin1. Scale bar: 20 μm. kd, knockdown.

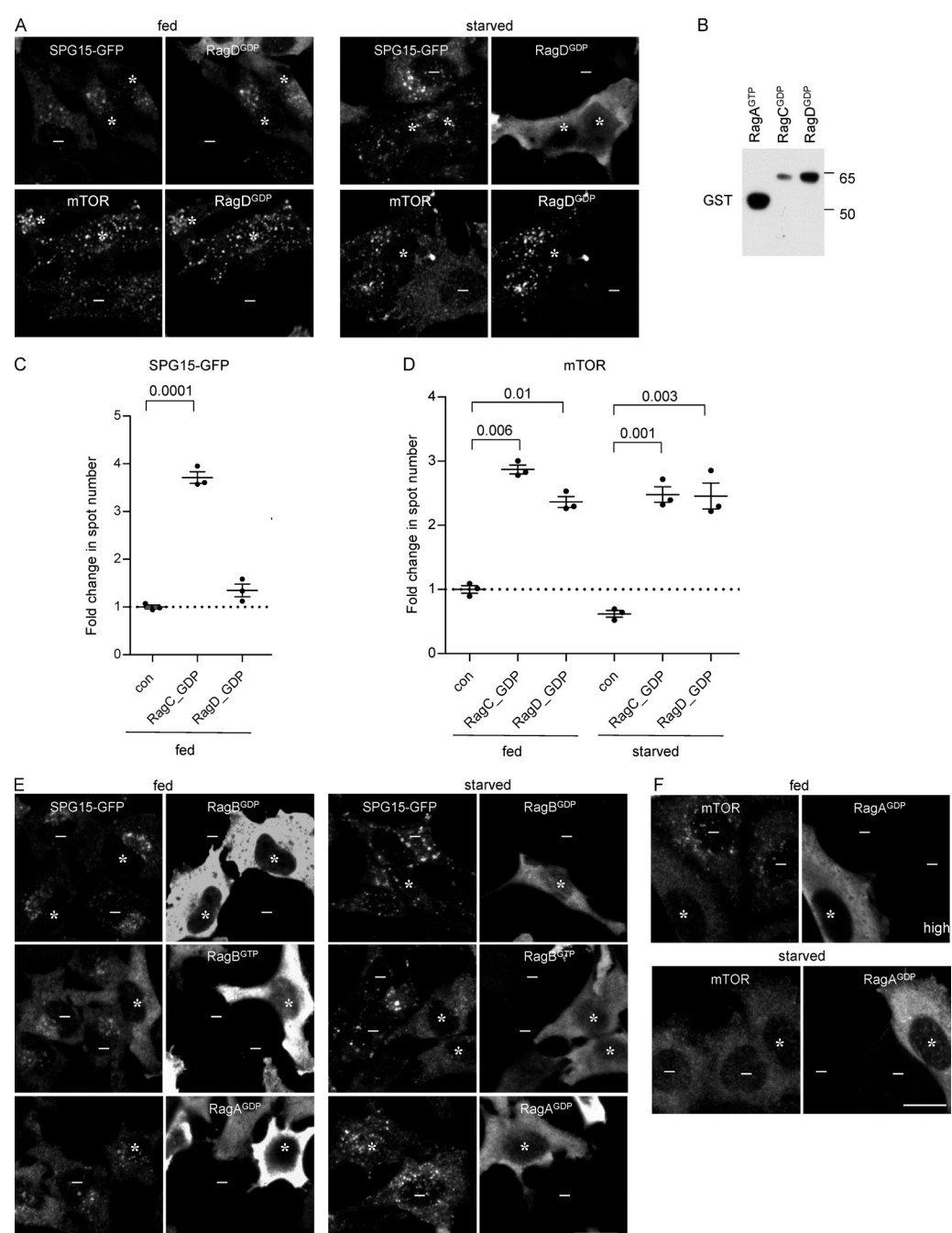

Figure S5. **Effect of Rags in different nucleotide loading states. (A)** IF labeling of SPG15-GFP (amplified with anti-GFP), mTOR, and HA-GST–tagged RagD[GDP] (detected using either anti-GST or anti-HA; all Rag constructs are tandem tagged). The RagD[GDP] did not have any appreciable effect on SPG15, but it caused an increase (compared with nonexpressing cells) in membrane-associated mTOR in starved cells. Asterisks denote RagD[GDP]-expressing cells, and minus signs denote RagD[GDP]-negative cells. Scale bar: 20 μm. **(B)** Western blots of whole-cell lysates from SPG15-GFP cells transiently transfected with HA-GST–tagged RagA or RagC constructs in their GDP and GTP locked forms and labeled with anti-GST to compare relative expression levels. IF showed similar transfection efficiencies. Image is representative of two independent experiments. **(C)** Quantification of the fold change in SPG15-GFP spot count per object in cells expressing either RagC[GDP] or RagD[GDP]. Unlike RagC[GDP], RagD[GDP] did not affect SPG15-GFP recruitment. The experiment was performed in biological triplicate (mean indicated; n = 3). The dotted line indicates no change. Error bars represent SEM. **(D)** Quantification of the fold change in mTOR spot count per object in cells expressing either RagC[GDP] or RagD[GDP], either fed or starved conditions (1 h). All data are normalized to nontransfected fed cells. RagD[GDP] caused an increase in membrane-associated mTOR in both fed and starved cells. The experiment was performed in biological triplicate (mean indicated; n = 3). The dotted line indicates no change. Error bars represent SEM. **(E)** IF labeling of SPG15-GFP (amplified with anti-GFP) in cells that had been transiently transfected with HA-GST-RagB in either GDP- or GTP-locked forms or with HA-GST-RagA in GDP-locked form. Where indicated, cells were incubated under fed or starved conditions (1 h). RagB[GTP] prevents the membrane association of SPG15. The asterisks indicate cells expressing the Rag construct, and the minus signs indicate cells not expressing the Rag construct. Scale bar: 20 μm. **(F)** IF double labeling of mTOR in cells that had been transiently transfected with HA-GST–tagged RagA[GDP]. Where indicated, cells were incubated under fed or starved conditions (1 h). Note that the overexpression of RagA[GDP] blocks recruitment of mTOR. The asterisks indicate cells expressing the Rag construct, and the minus signs indicate cells not expressing the Rag construct. Scale bar: 20 μm.

**Table S1, provided online, lists interactions identified by BioID.**

