## [Peer Review File · The Journal of Cell Biology]

Rag GTPases and Phosphatidylinositol 3-Phosphate Mediate Recruitment of the AP-5/SPG11/SPG15 Complex

Jennifer Hirst, Geoffrey Hesketh, Anne-Claude Gingras, and Margaret Robinson

Corresponding Author(s): Margaret Robinson, University of Cambridge and Jennifer Hirst, University of Cambridge

Review Timeline:

Submission Date:	2020-02-13
Editorial Decision:	2020-03-25
Revision Received:	2020-05-21
Editorial Decision:	2020-06-11
Revision Received:	2020-09-25
Editorial Decision:	2020-10-23
Revision Received:	2020-11-23

Monitoring Editor: Scott Emr

Scientific Editor: Tim Spencer

Transaction Report:

DOI: <https://doi.org/10.1083/jcb.202002075>

forwarding the first invite to revise for the Margaret Robinson paper for your reference:

March 25, 2020

Re: JCB manuscript #202002075

Prof. Margaret S Robinson
University of Cambridge
CIMR
Cambridge CB2 0XY
United Kingdom

Dear Prof. Robinson,

Thank you for submitting your manuscript entitled "Rag GTPases and Phosphatidylinositol 3-Phosphate Mediate Recruitment of the AP-5/SPG11/SPG15 Complex". The manuscript was assessed by expert reviewers, whose comments are appended to this letter. We invite you to submit a revision if you can address the reviewers' key concerns, as outlined here.

Your manuscript has received mainly positive reviews. The reviewers do, however, request a reasonable set of revisions prior to acceptance. Please pay particular attention to the comments in Rev#1's review (you do not need to address the comments from Rev#3 that recommend providing more mechanistic insight into the role of nutrient signaling). As pointed out by Rev#1, the manuscript needs biochemical evidence for the interaction between RagC-GDP and the SPG11/15 complex.

GENERAL GUIDELINES:

Text limits: Character count for an Article is < 40,000, not including spaces. Count includes title page, abstract, introduction, results, discussion, acknowledgments, and figure legends. Count does not include materials and methods, references, tables, or supplemental legends.

Figures: Articles may have up to 10 main text figures. Figures must be prepared according to the policies outlined in our Instructions to Authors, under Data Presentation, <http://jcb.rupress.org/site/misc/ifora.xhtml>. All figures in accepted manuscripts will be screened prior to publication.

IMPORTANT: It is JCB policy that if requested, original data images must be made available. Failure to provide original images upon request will result in unavoidable delays in publication. Please ensure that you have access to all original microscopy and blot data images before submitting your revision.

Supplemental information: There are strict limits on the allowable amount of supplemental data.

Articles may have up to 5 supplemental figures. Up to 10 supplemental videos or flash animations are allowed. A summary of all supplemental material should appear at the end of the Materials and methods section.

The typical timeframe for revisions is three months; if submitted within this timeframe, novelty will not be reassessed at the final decision - this can be extended given the current pandemic. Please note that papers are generally considered through only one revision cycle, so any revised manuscript will likely be either accepted or rejected.

Thank you for this interesting contribution to Journal of Cell Biology. You can contact us at the journal office with any questions, cellbio@rockefeller.edu or call (212) 327-8588.

Sincerely,

Scott Emr, Ph.D.
Monitoring Editor

Marie Anne O'Donnell, Ph.D.
Scientific Editor

Journal of Cell Biology

Reviewer #1 (Comments to the Authors (Required)):

In this manuscript, Hirst and Robinson document the recruitment of the mysterious AP-5 complex onto late endosomes/ lysosomes via interactions of the associated SPG11/SPG15 complex with both phosphatidylinositol-3-phosphate (PtdIns3P) and the Rag GTPase, RagC. The Rag GTPases have been previously implicated only in the recruitment of mTORC1 to lysosomes, and PtdIns3P is thought to be largely limited to early/ maturing endosomes. The authors first show that mutagenesis of the FYVE domain from SPG15 to disrupt PtdIns3P-binding blocks recruitment of full-length SPG15 to lysosomes - suggesting that a small pool of PtdIns3P must exist on lysosomes and is required for recruitment of the complex. Similarly, knockdown of RagC but not of other lysosomal GTPases impairs SPG15 recruitment, whereas overexpression of the GDP-locked form of RagC but not of other Rag GTPases enhances recruitment. Antithetically, starvation (usually associated with GTP-locked RagC) promotes SPG15 recruitment in a RagC-dependent manner. Limited knockdown data support the notion that recruitment is entirely mediated by the SPG11/15 complex and not by AP-5. The authors conclude that recruitment of SPG11/15 to lysosomes depends on coincidence detection of both PtdIns3P and RagC.

The manuscript addresses an important question in the field in defining the molecular determinants of AP-5 association with membranes, and the finding that RagC and PtdIns3P have something to do with this recruitment is very exciting for the field. In particular, the data provide the first evidence

of Rag-dependent recruitment of a presumed trafficking protein, and support a frequently made supposition that PtdIns3P on structures other than early endosomes plays an important role in membrane dynamics. With some exceptions noted below, the data are largely solid and quantitatively assessed. The main conclusions are well supported by functional data. Thus, in principle this manuscript would be of interest to the readers of the JCB. However, there are a number of concerns that need to be addressed.

The main concern is that all of the data employ cellular manipulations and assessment of SPG15, SPG11, or AP-5 by fluorescence/ immunofluorescence microscopy; there are no in vitro data or biochemical approaches to validate and extend the conclusions. In particular, the manuscript would benefit from biochemical evidence of an interaction - direct or not - between RagC-GDP and the SPG11/15 complex. Pull-downs in Suppl. Fig. 6B fail to detect binding of the AP-5 subunit AP5Z1, but neither SPG11 nor SPG15 were probed. It seems to me that this could easily be tested even if specific blotting antibodies for SPG11 or SPG15 were unavailable by exploiting the stable cell lines expressing GFP-SGP11 or -SPG15. One also wonders whether it might be possible to reconstitute the recruitment of tagged AP5/SPG11/SPG15 from cell lysates onto PtdIns3P-containing liposomes in the presence of recombinant RagC or RagC/RagA complex. Absent data such as this, the impact of knocking down SPG11 but not AP-5 on SPG15 recruitment in Figure 6B-D is weak evidence that this complex mediates the RagC-GDP interaction, and all of the remaining data could be potentially explained by indirect interactions of RagC-associated factors with SPG11/15/AP-5. Even the requirement of the SPG15-FYVE domain interaction with PtdIns3P for AP-5 recruitment is not validated by data presented in the paper (see point 8 below).

In addition to this main issue, there are several other more easily addressed concerns as detailed below.

1.) In general, many of the images shown are not particularly convincing of the changes in membrane recruitment described in the text and the quantification, particularly recruitment of SPG15/11/AP-5 in fed cells. There is substantial variation among the images in the level of recruitment and the number of puncta shown, particularly in Figures 4-6 and their Supplemental associated figures, and the images often do not match the quantification or the conclusions drawn in the text. A particularly egregious example is Suppl. Fig. 6A, which supposedly shows RagC-GDP-dependent stabilization of AP5Z1-GFP on membranes, but barely any puncta can be seen. The authors would benefit from ensuring that the most optimal images are shown, that the images are optimized (uniformly within a figure) for brightness/ contrast, and that the images match the quantification. It would also benefit readers if the authors explicitly stated in the Results and/or Figure legends why some images show perinuclear accumulation whereas others show more peripheral accumulation (these fit the biology of the knockdowns).

2.) All of the knockdowns employ only a single type of siRNA treatment (the Methods are a bit confusing, but they appear to be siRNA pools), and no rescue experiments are performed. The key experiments showing positive results (e.g. RagC knockdown) should be repeated with at least two separate individual siRNAs. Optimally rescue with an siRNA-resistant expression construct should also be done.

3.) For Figure 1 and Suppl. Fig. 1, the cell lines used and the origin of the GFP signal are not clear. Are these stably transduced HeLa cells in Fig. 1, S1A and S1C? It is indicated that the SPG15-GFP is expressed under its own promoter in some of these, but how? Are these knock-ins, or were the cells transfected with a construct containing a fragment of the SPG15 gene with GFP inserted in frame? These points need to be clarified in the text of the Results (briefly), Materials and Methods

(extensively, and the Figure legend (briefly). Additionally, Suppl. Fig. S1A and B would benefit from the addition of highly magnified insets to better demonstrate overlap (or not) with LAMP1 and EEA1.

4.) In Figure 2 and Suppl. Fig. 2, the following concerns should be addressed.

a.) In panel A, are all of the images taken at the same brightness/ contrast and similarly manipulated post-acquisition? The signal for SPG15-GFP in cells treated with wortmannin seems much higher than the others.

b.) In panel B, it would be helpful if it were indicated on the Figure itself that the values correspond to SPG15-GFP and not to LAMP1.

c.) In the text describing the results, I suggest that the authors refrain from speculation on why the spots are bigger upon treatment with the PIKfyve inhibitor until the Discussion. While the explanation posited is plausible, given that PtdIns(3,5)P₂ activates TRPML and TPC channels on lysosomes that are required for lysosomal metabolic maintenance, it is equally (or perhaps more) likely that PIKfyve inhibition results in indirect inhibition of mTORC1 which then leads to increased AP-5 recruitment.

d.) In Suppl. Fig. 2B, the PI5K inhibitor treatments need to be compared to controls without the inhibitor.

e.) The Results at bottom of page 8 cite Figure 2C as indicative that GFP-FYVE(SP15) is expressed at near endogenous levels, but no evidence (at least here, and I could not find it in the cited references) is provided that the full-length GFP-SP15 is expressed at endogenous levels. Either provide the proof or remove the statement.

f.) Panels in Figure 2E would benefit from highly magnified insets to emphasize the colocalization or lack thereof.

5.) In Figure 3F, how many cells/ profiles in how many experiments were quantified?

6.) Figure 4 and Suppl. Fig. S4 would benefit from quantification of # of puncta/ cell as in previous figures. Alternatively, membrane association of AP-5 in the cell population could be assessed biochemically by cell fractionation to membrane and cytosol fractions.

7.) In Suppl. Fig. S5A, is the lower left panel of the fed cells mTOR in the same cells as those shown to the right?

8.) On page 14 in the Results, the authors state: "The PI3P interaction [of the AP-5/SPG11/15 complex] is clearly mediated by the SPG15 FYVE domain", but this is not supported by the data shown - all that is shown is that localization of SPG15 itself is ablated by a mutation predicted to interfere with FYVE binding to PtdIns3P. Indeed, it would be a good experiment to express either the intact or R183A mutant form of HA-SPG15 in cells knocked down for endogenous SPG15 and expressing SPG11-GFP or tagged AP-5 subunits to test whether this statement is true. In the absence of such data, the conclusion cannot be drawn.

9.) The observation that overexpression of RagC-GDP but not of RagD-GDP induces SPG15 recruitment is interesting; would this not provide a nice potential approach to identify a potential interacting region of RagC for the complex through domain swaps between RagC and RagD?

10.) The authors provide a rather unsatisfying discussion of the paradoxical recruitment of SPG15 to lysosomes by starvation but also by overexpression of RagC-GDP - confusing given that starvation is thought to induce GTP exchange onto RagC. Might this imply that the SPG11/15/AP-5 complex is recruited at a time prior to full activation of the Rags (e.g. by GEF-mediated activation of

RagA/B) but after partial activation through GTPase activity on RagC?

11.) More details are needed in the Materials and Methods for the quantification of particle size and number in Figures 2, 3, 5, 6 and S5.

Reviewer #2 (Comments to the Authors (Required)):

This is an interesting study that provides novel information about recruitment of AP-5/SPG11/SPG15 Complexes to late endosomes/lysosomes. The involvement of the RAG complexes and the unexpected effects of their GTP/GDP binding is novel and will certainly lead to further work by several laboratories. The imaging and colocalization studies were generally of exceptional quality and give confidence in the experimental results that are described.

Comments

1. In Figure 3A the Arl18 knockdown does look like it is causing a redistribution of SPG15-GFP.
2. The interpretation of Figure 6C is not clear from the images shown but seems clear in the analysis of many cells. If the analysis is correct, maybe different fields should be shown in 6C.

Reviewer #3 (Comments to the Authors (Required)):

The manuscript by Hirst and Robinson reports that AP-5 and its interaction partners SPG11 and SPG15 are recruited onto late endosomes and lysosomes by coincidence detection of Rag GTPases and the phosphoinositide PI3P. Interaction with PI3P was found to be via the FYVE domain of SPG15, which by its own localized to endosomes. GDP-locked RagC promoted recruitment of the AP-5/SPG11/SPG15 complex to endosomes/lysosomes, whereas GTP-locked RagA inhibited recruitment. Localization of the complex to late endosomes/lysosomes was enhanced by amino acid starvation, and the authors speculate that the AP-5/SPG11/SPG15 complex could play a role in starvation signaling. The conclusions of this manuscript are well supported by data of good quality, and this could be the beginning of a very interesting story. However, by themselves the present findings represent a rather limited advance and do not merit publication in JCB.

Major points:

1. The major deficit of this manuscript is the complete absence of functional data. The authors have previously assigned a role for the AP-5/SPG11/SPG15 complex in protein sorting, and now they speculate that it could be involved in nutrient signaling. However, in the absence of data that show the involvement of AP-5/SPG11/SPG15 in nutrient signaling, that this depends on PI3P/Rag coincidence detection, and that it has consequences for the cell, the impact of the present findings is moderate.
2. The authors have previously shown that wortmannin prevents recruitment of the AP-5/SPG11/SPG15 complex to endosomes (Hirst et al., MBC, 2013), so the aspect of PI3P dependence is not novel. Additionally it is not clear why the authors did not choose to use more subclass specific PI3K inhibitors than wortmannin.

3. The coincidence detection of PI3P and Rags by SPG15 is novel and interesting, but further molecular insight is required. Whereas the FYVE domain of SPG15 clearly recognizes PI3P, the authors should use deletion mutagenesis to establish which domain binds Rags.

Minor points:

1. It is difficult to understand how SPG15 could be recruited to endosomes/lysosomes by coincidence detection of PI3P and Rags as long as the FYVE domain alone of SPG15 is sufficient to localize to endosomes. Do the authors think that there is allosteric regulation of AP-5/SPG1/SPG15 interaction with endosomes?

2. "Starvation" could mean many things, and I could not find how it was actually done. This should be mentioned explicitly in the Results, with details in Figure legend or MM.

Margaret S. Robinson
Emeritus Professor of Molecular Cell Biology

The Journal of Cell Biology

21 May, 2020

Dear Editors,

Thank you for the referees' comments on our manuscript, "Rag GTPases and Phosphatidylinositol 3-Phosphate Mediate Recruitment of the AP-5/SPG11/SPG15 Complex" (202002075). We were very happy to hear that our manuscript had received mainly positive reviews. Because of the current pandemic, our lab is closed for the foreseeable future, so we don't know when we might be able to do the additional experiments suggested by the referees. Even if our lab were to reopen within the next month or so (which is unlikely), Jenny Hirst, who carried out all of the experimental work in the paper, is on full time secondment at a COVID-19 testing facility for the next three months. (Increasing the testing capacity is particularly important in the UK, as until recently we have had one of the lowest numbers of tests per capita in Europe, as well as the highest number of COVID-19-related deaths.) Therefore, we hope that because of their generally encouraging comments, the reviewers will accept changes that were made mainly by rewriting parts of the text. We have been able to address all of their concerns, and in addition to the textual changes, we have some unpublished results, in particular from a colleague's lab, which answer several of their questions. As you advised, we are paying particular attention to the points raised by Reviewer 1.

From Reviewer 1:

The main concern is that all of the data employ cellular manipulations and assessment of SPG15, SPG11, or AP-5 by fluorescence/ immunofluorescence microscopy; there are no in vitro data or biochemical approaches to validate and extend the conclusions. In particular, the manuscript would benefit from biochemical evidence of an interaction - direct or not - between RagC-GDP and the SPG11/15 complex. Pull-downs in Suppl. Fig. 6B fail to detect binding of the AP-5 subunit AP5Z1, but neither SPG11 nor SPG15 were probed. It seems to me that this could easily be tested even if specific blotting antibodies for SPG11 or SPG15 were unavailable by exploiting the stable cell lines expressing GFP-SGP11 or -SPG15. One also

wonders whether it might be possible to reconstitute the recruitment of tagged AP5/SPG11/SPG15 from cell lysates onto PtdIns3P-containing liposomes in the presence of recombinant RagC or RagC/RagA complex. Absent data such as this, the impact of knocking down SPG11 but not AP-5 on SPG15 recruitment in Figure 6B-D is weak evidence that this complex mediates the RagC-GDP interaction, and all of the remaining data could be potentially explained by indirect interactions of RagC-associated factors with SPG11/15/AP-5. Even the requirement of the SPG15-FYVE domain interaction with PtdIns3P for AP-5 recruitment is not validated by data presented in the paper (see point 8 below).

We completely agree that biochemical data would strengthen our conclusion that SPG11/SPG15 interacts either directly or indirectly with GDP-bound RagC. The reviewer suggests that we try carrying out our Rag pulldowns (shown in our original Supplemental Fig. 6B but now deleted; see our response to point 1 from Reviewer 1) on cells expressing GFP-tagged constructs, and then probe blots with anti-GFP. We did in fact do this experiment, but did not see any specific labelling, and we now mention that in the text (pages 13-14). Our inability to pull down AP-5/SPG11/SPG15 with Rags actually fits in well with what we know about other types of coats and the membrane-associated proteins that recruit them. These interactions tend to be both weak and conformation-dependent, causing the proteins to dissociate when cells are broken open. The use of detergents is also problematic, when the interactions depend at least in part on lipids. For instance, to the best of our knowledge, nobody has been able to co-immunoprecipitate APs or COPI with ARF1, even though the proteins interact on membranes and ARF1 is essential for the recruitment of all of these coats.

An approach that often works well for identifying transient interactions is proximity biotinylation, or BioID. This is a method carried out in living cells, in which a “bait” protein is fused to a promiscuous biotin ligase, BirA*. Upon addition of biotin to the medium, proteins that interact with and/or are in close proximity to the bait become biotinylated, and can be pulled down using streptavidin. The reason we did not try this approach ourselves was that we knew that a former colleague at the CIMR, Dr Geoffrey Hesketh, was carrying out proximity biotinylation experiments in the lab of Dr Anne-Claude Gingras, but on a much larger scale and using more powerful mass spectrometry facilities than our own. Their unpublished results strongly support our hypothesis, and they have generously given us permission to show their findings to the reviewers. The image on the next page, provided by Geoff Hesketh, shows some of the hits from his BioID experiments carried out using various BirA*-tagged bait constructs, including all four of the Rag GTPases, either wild-type, GTP-locked, or GDP-locked.

SPG15 and SPG11 are in the top two rows, indicated with a dotted box. Note how they are specifically labelled by GDP-locked Rags, rather than wild-type or GTP-locked. This all fits in remarkably well with the model we propose in Figure 7, which we had already formulated before we saw the BioID data. In our original manuscript, we acknowledged Geoff Hesketh for

reading the manuscript and for helpful discussions, but we have now included him as a co-author in light of his exceptional intellectual input into our study. However, this figure is for the referees only, because he and Anne-Claude Gingras are planning to publish it as part of a larger story of their own.

Regarding whether or not the interaction is direct, we agree that this is a very important question, and one to which we do not yet have an answer. Therefore, we were very careful in the paper to make it clear that the Rag interaction could be mediated by another unidentified protein(s). We had already mentioned in the discussion that liposome binding experiments would probably be the best way to determine whether or not the interaction is direct, and we are planning to carry out these experiments in collaboration with our structural biology colleague, Dr Lauren Jackson at Vanderbilt University. However, these are complicated and time-consuming experiments that we feel are beyond the scope of the present study, especially because both of our labs are currently in lockdown.

The first referee also had some more minor comments:

1.) *In general, many of the images shown are not particularly convincing of the changes in membrane recruitment described in the text and the quantification, particularly recruitment of*

SPG15/11/AP-5 in fed cells. There is substantial variation among the images in the level of recruitment and the number of puncta shown, particularly in Figures 4-6 and their Supplemental associated figures, and the images often do not match the quantification or the conclusions drawn in the text. A particularly egregious example is Suppl. Fig. 6A, which supposedly shows RagC-GDP-dependent stabilization of AP5Z1-GFP on membranes, but barely any puncta can be seen. The authors would benefit from ensuring that the most optimal images are shown, that the images are optimized (uniformly within a figure) for brightness/contrast, and that the images match the quantification. It would also benefit readers if the authors explicitly stated in the Results and/or Figure legends why some images show perinuclear accumulation whereas others show more peripheral accumulation (these fit the biology of the knockdowns).

We take the reviewer's point that there is variation amongst the images in the level of recruitment and number of puncta. The problem is that despite our best attempts, the cell lines are heterogeneous. All of the cells were sorted by flow cytometry and then clonal lines were isolated, and these clonal lines have been re-sorted, but there is still a lot of variability between cells. This is why the quantification was so important, because it allowed us to objectively sample many cells in three separate experiments for each condition. So the quantitative results are really much more informative than the individual images. We now make a point about heterogeneity early on in the results (page 7). Regarding Supplemental Figure 6A, which shows AP5Z1-GFP cells, this cell line has always been the most problematic, and because these are fed cells, the discrete puncta can really only be seen in the RagC^{GDP}-expressing cells, where they colocalise with RagC^{GDP}. But because we are one supplemental figure over the limit, we have decided to delete Supplemental Figure 6 entirely. Regarding optimisation for brightness and contrast, we were very careful to process all images the same way (i.e., same exposure time and same manipulations), as we believe this is the only way they can be compared objectively, either by eye or electronically. This has now been clarified in the Materials and Methods (page 24). The reviewer also points out that some images show perinuclear accumulation while others show more peripheral accumulation, and that this is consistent with the biology of the knockdowns. That is absolutely right, and we didn't mention it because we were trying to keep our focus on membrane association. However, we now make this point in the legend for Figure 3A (page 32).

2.) All of the knockdowns employ only a single type of siRNA treatment (the Methods are a bit confusing, but they appear to be siRNA pools), and no rescue experiments are performed. The key experiments showing positive results (e.g. RagC knockdown) should be repeated with at

least two separate individual siRNAs. Optimally rescue with an siRNA-resistant expression construct should also be done.

We agree that ideally one should carry out knockdowns using more than one siRNAs and/or include a rescue experiment. However, in this particular instance we obtained exactly the same phenotype when we knocked down Ragulator, which acts upstream from the Rags, and a similar (albeit weaker) phenotype when we knocked down RagA, which acts together with RagC. The fact that we got similar phenotypes when knocking down three components of the same pathway provides strong evidence that these effects are not off-target.

3.) For Figure 1 and Suppl. Fig. 1, the cell lines used and the origin of the GFP signal are not clear. Are these stably transduced HeLa cells in Fig. 1, S1A and S1C? It is indicated that the SPG15-GFP is expressed under its own promoter in some of these, but how? Are these knock-ins, or were the cells transfected with a construct containing a fragment of the SPG15 gene with GFP inserted in frame? These points need to be clarified in the text of the Results (briefly), Materials and Methods (extensively, and the Figure legend (briefly). Additionally, Suppl. Fig. S1A and B would benefit from the addition of highly magnified insets to better demonstrate overlap (or not) with LAMP1 and EEA1.

We take the referee's point that the origins of these cell lines could have been more clearly explained. They were first published 10 years ago by Slabicki et al., who then generously shared them with us, and we say all this in the Materials and Methods. However, we agree that it would be helpful to say more about how the cells were actually made. They were generated using "BAC TransgeneOmics", a technique that was being developed at the time in Tony Hyman's and Frank Buchholz's labs in Dresden, both of whom were co-authors on the Slabicki et al. paper. We now go into more detail about the origin of the cells, in the Results (page 7), Materials and Methods (page 21), and legend to Figure 1. We also explain that in spite of cloning and re-sorting, the cells are heterogeneous (pages 7 and 21), and we include magnified inserts in Supplemental Figure 1.

4.) In Figure 2 and Suppl. Fig. 2, the following concerns should be addressed.

a.) In panel A, are all of the images taken at the same brightness/ contrast and similarly manipulated postacquisition? The signal for SPG15-GFP in cells treated with wortmannin seems much higher than the others.

Yes, as we explain above in the answer to Question 1, the images were always taken at the same brightness and contrast, but some of the cells express more SPG15-GFP than others, which is one of the reasons quantification was so important. The reason for the brighter signal in the cells treated with wortmannin is that most of the construct was cytosolic. Similarly, there is more of a cytosolic signal in fed cells than in starved cells or cells treated with a PI 5-kinase inhibitor, because there is less AP-5/SPG11/SPG15 on the membrane. We now make this clear in the figure legend (page 30).

b.) In panel B, it would be helpful if it were indicated on the Figure itself that the values correspond to SPG15- GFP and not to LAMP1.

This has been done.

c.) In the text describing the results, I suggest that the authors refrain from speculation on why the spots are bigger upon treatment with the PIKfyve inhibitor until the Discussion. While the explanation posited is plausible, given that PtdIns(3,5)P2 activates TRPML and TPC channels on lysosomes that are required for lysosomal metabolic maintenance, it is equally (or perhaps more) likely that PIKfyve inhibition results in indirect inhibition of mTORC1 which then leads to increased AP-5 recruitment.

We thank the referee for the suggestion, and have removed the speculation from the Results section (page 8). In the figure legend, we now mention that some of the effects of PI5K inhibition may be indirect (page 31).

d.) In Suppl. Fig. 2B, the PI5K inhibitor treatments need to be compared to controls without the inhibitor.

The figure has now been expanded to include controls without the inhibitor.

e.) The Results at bottom of page 8 cite Figure 2C as indicative that GFP-FYVE(SP15) is expressed at near endogenous levels, but no evidence (at least here, and I could not find it in the cited references) is provided that the full-length GFP-SP15 is expressed at endogenous levels. Either provide the proof or remove the statement.

The reason we said that GFP-FYVE(SP15) is expressed at near-endogenous levels is that Figure 2C shows that it is expressed at similar levels to the full-length GFP-SP15, and the

full-length construct makes use of its endogenous promoter and was shown to be expressed at near-endogenous levels in the first paper that described this construct, Słabicki et al. 2010. We now simply say that the GFP-FYVE(SPG15) construct was expressed at similar levels to the full-length construct (page 8).

f.) Panels in Figure 2E would benefit from highly magnified insets to emphasize the colocalization or lack thereof.

This has been done.

5.) In Figure 3F, how many cells/ profiles in how many experiments were quantified?

As we explain in the Materials and Methods (page 24), the cells were always quantified in the same way: a minimum of 20 cells were analysed for each condition and repeated at least three times. We now added that information to the legend for Figure 3F.

6.) Figure 4 and Suppl. Fig. S4 would benefit from quantification of # of puncta/ cell as in previous figures. Alternatively, membrane association of AP-5 in the cell population could be assessed biochemically by cell fractionation to membrane and cytosol fractions.

We did try to quantify the number of puncta per cell for the NPRL3 and FLCN knockdowns, even though no difference was discernable by eye, but the data were inconclusive. In the end, we decided to quantify only those conditions where there was a robust difference. We also tested – for several of our conditions – whether changes in localisation could be correlated with changes in the soluble pool, as determined by homogenisation and centrifugation, exactly as suggested by the reviewer. However, even in wortmannin-treated cells, there was no clear difference in the amount of AP-5/SPG11/SPG15 in the soluble fraction. This may have to do with the behaviour of the complex in broken-open cells. A parallel situation arose nearly 30 years ago, when we tried and failed to find differences in the soluble fraction of AP-1 and COPI in brefeldin A-treated cells (see Robinson and Kreis, 1992, PMID 1555237), even though the drug caused both types of coats to be lost from membranes.

7.) In Suppl. Fig. S5A, is the lower left panel of the fed cells mTOR in the same cells as those shown to the right?

Yes, we have now added the missing label and thank the referee for pointing this out

8.) *On page 14 in the Results, the authors state: "The PI3P interaction [of the AP-5/SPG11/15 complex] is clearly mediated by the SPG15 FYVE domain", but this is not supported by the data shown - all that is shown is that localization of SPG15 itself is ablated by a mutation predicted to interfere with FYVE binding to PtdIns3P. Indeed, it would be a good experiment to express either the intact or R183A mutant form of HA-SPG15 in cells knocked down for endogenous SPG15 and expressing SPG11-GFP or tagged AP-5 subunits to test whether this statement is true. In the absence of such data, the conclusion cannot be drawn.*

Although we have not tried to localise the other subunits in cells expressing the R183A mutant form of SPG15, we do have evidence from earlier studies that wortmannin causes the loss of tagged μ 5 and σ 5 from the membrane. We have now toned down the sentence to say, "The PI3P interaction is a function of the SPG15 FYVE domain" (page 14).

9.) *The observation that overexpression of RagC-GDP but not of RagD-GDP induces SPG15 recruitment is interesting; would this not provide a nice potential approach to identify a potential interacting region of RagC for the complex through domain swaps between RagC and RagD?*

This is something we very much want to pursue, and these experiments were underway when our lab got shut down on March 20th. Our first concern was that there might be a more trivial explanation for the difference between RagC^{GDP} and RagD^{GDP}, because the two constructs, which were kindly provided by the Sabatini lab, were not entirely equivalent. There is a highly conserved stretch of residues containing the P loop, which plays a key role in nucleotide binding, and which is identical in RagC and RagD: GLRRGK**SSI**. Both Rags had serine-to-leucine substitutions, but the sequence was GLRRGK**LSI** in RagC and GLRRGK**SLI** in RagD. One of our colleagues, who had carried out structural studies on the Rags, told us that the first serine was more important for nucleotide binding, so we made the GLRRGK**LSI** mutation in RagD. We obtained exactly the same results, with the RagD mutant promoting mTOR recruitment but not SPG15-GFP recruitment, so there really is something different about the two Rags. We are very keen to make chimeras when our lab reopens.

10.) *The authors provide a rather unsatisfying discussion of the paradoxical recruitment of SPG15 to lysosomes by starvation but also by overexpression of RagC-GDP - confusing given that starvation is thought to induce GTP exchange onto RagC. Might this imply that the*

SPG11/15/AP-5 complex is recruited at a time prior to full activation of the Rags (e.g. by GEF-mediated activation of RagA/B) but after partial activation through GTPase activity on RagC?

The referee makes an important point here. Although it is tempting to speculate that the molecular mechanism for increased recruitment of SPG15 onto membranes is always the same, the similar effects of starvation and overexpression of RagC^{GDP} are indeed paradoxical. We were careful not to imply that the mechanisms were necessarily the same, and our Discussion we included the sentence “Precisely how this apparent preference of the AP-5/SPG11/SPG15 complex for both Rags in their GDP-bound state fits in with its increased recruitment during starvation is still unclear.” The referee’s suggestion is one that we had considered and we now include it on page 16.

11.) More details are needed in the Materials and Methods for the quantification of particle size and number in Figures 2, 3, 5, 6 and S5.

We now include these details in the Materials and Methods section (page 24).

Reviewer 2:

1. In Figure 3A the Arl18 knockdown does look like it is causing a redistribution of SPG15-GFP.

Reviewer 1 also commented on how some knockdowns cause increased perinuclear labelling while others cause increased peripheral labelling, and Arl8 is a good example of a knockdown that causes an increase in perinuclear labelling. Although we wanted to keep our focus on how much of the SPG15-GFP was membrane-associated, rather than where those membranes were, we now acknowledge that the Arl8 knockdown causes a redistribution of SPG15-GFP in the figure legend (page 32).

2. The interpretation of Figure 6C is not clear from the images shown but seems clear in the analysis of many cells. If the analysis is correct, maybe different fields should be shown in 6C.

Although there is still some weak punctate labeling in the SPG11 knockdown cells in Figure 6C, this is consistent with the graph in Figure 6D, which shows that the spot number is reduced but does not go down to zero. What is clearer from the immunofluorescence image is

that the relative amount of cytosolic labelling is increased, and we now make this point in the figure legend.

Reviewer 3:

1. The major deficit of this manuscript is the complete absence of functional data. The authors have previously assigned a role for the AP-5/SPG11/SPG15 complex in protein sorting, and now they speculate that it could be involved in nutrient signaling. However, in the absence of data that show the involvement of AP-5/SPG11/SPG15 in nutrient signaling, that this depends on PI3P/Rag coincidence detection, and that it has consequences for the cell, the impact of the present findings is moderate.

We agree that it will be important to investigate the potential role of the AP-5/SPG11/SPG15 complex in nutrient signalling. In fact, we had some encouraging preliminary results when our lab was shut down, and we plan to make this the focus of a follow-up study.

2. The authors have previously shown that wortmannin prevents recruitment of the AP-5/SPG11/SPG15 complex to endosomes (Hirst et al., MBC, 2013), so the aspect of PI3P dependence is not novel. Additionally it is not clear why the authors did not choose to use more subclass specific PI3K inhibitors than wortmannin.

We realise that the wortmannin result is not novel, but we have followed it up by making mutations in the SPG15 FYVE domain that prevent PI3P binding, which we feel is a more rigorous test of PI3P dependence than investigating other inhibitors. Moreover, our finding that the SPG15 FYVE domain on its own goes to early endosomes rather than late endosomes is completely novel, and an important extension of our previous findings.

3. The coincidence detection of PI3P and Rags by SPG15 is novel and interesting, but further molecular insight is required. Whereas the FYVE domain of SPG15 clearly recognizes PI3P, the authors should use deletion mutagenesis to establish which domain binds Rags.

This is something we intend to do, but SPG11 and SPG15 are both very large proteins of ~280 kD, and the only clear domains are the FYVE domain in SPG15 and the putative β -propeller in SPG11. Otherwise, the two proteins are almost entirely α -solenoid. Based on other large α -solenoid-containing proteins involved in membrane traffic (e.g., clathrin heavy chain, COPI α and β' subunits, and Sec31), we suspect that the α -solenoids of SPG11 and SPG15 interact

extensively with each to form an obligatory heterodimer. Thus, deletion mutants are likely to be non-functional. Nevertheless, we did try to make several truncations of SPG15; however, all of these constructs were cytosolic. Probably the only way to establish how the SPG11/SPG15 complex binds to Rags would be to express the entire complex and carry out structural studies. This is something we plan to do in collaboration with Lauren Jackson's lab, but it will be a major undertaking that is likely to take years rather than months.

(Minor points)

1. It is difficult to understand how SPG15 could be recruited to endosomes/lysosomes by coincidence detection of PI3P and Rags as long as the FYVE domain alone of SPG15 is sufficient to localize to endosomes. Do the authors think that there is allosteric regulation of AP-5/SPG1/SPG15 interaction with endosomes?

The reviewer makes an interesting and important point here: the PI3P interaction is sufficient to target the SPG15 FYVE domain on its own to membranes, but not in the context of the whole complex. There are precedents for coincidence detection involving allosteric regulation; probably the best characterised of these is the conformational change that occurs in AP-2 when it comes in contact with the plasma membrane, causing binding sites for PIP2, cargo, and clathrin to be exposed. We now discuss this in the text (pages 18-19).

2. "Starvation" could mean many things, and I could not find how it was actually done. This should be mentioned explicitly in the Results, with details in Figure legend or MM.

We now specify how we did the starvation in the Results section (page 8) and the first figure legend (we actually did specify how we did it in the Materials and Methods, on page 22).

Finally, we thank the reviewers for their constructive suggestions, which we feel have improved our manuscript. We hope that with the changes we have made, the referees will recommend publication. Thank you again for considering our manuscript, and we look forward to hearing from you

Yours sincerely,

Margaret S. Robinson

Jennifer Hirst

CIMR, Cambridge CB2 0XY, UK
Email: msr12@cam.ac.uk, jh228@cam.ac.uk

June 11, 2020

Re: JCB manuscript #202002075R

Prof. Margaret S Robinson
University of Cambridge
CIMR
Cambridge CB2 0XY
United Kingdom

Dear Prof. Robinson,

Thank you for submitting your manuscript entitled "Rag GTPases and Phosphatidylinositol 3-Phosphate Mediate Recruitment of the AP-5/SPG11/SPG15 Complex"

We fully appreciate the difficulty of conducting experiments requested by reviewers due to the pandemic and the closure of research labs in the UK. However, we feel the reviewers' comments (especially those of reviewers 1 and 3) are reasonable and appropriate given this is not the first paper addressing the localization and function of AP-5. Therefore, we think it necessary to complete the additional experimental work to address the major concerns raised by Reviews 1 and 3 regarding validation of the interaction between RagC and AP-5, as well as the potential role of AP-5 in regulating nutrient signaling, and are happy to extend the revision period.

GENERAL GUIDELINES:

Text limits: Character count for an Article is < 40,000, not including spaces. Count includes title page, abstract, introduction, results, discussion, acknowledgments, and figure legends. Count does not include materials and methods, references, tables, or supplemental legends.

Figures: Articles may have up to 10 main text figures. Figures must be prepared according to the policies outlined in our Instructions to Authors, under Data Presentation, <http://jcb.rupress.org/site/misc/ifora.xhtml>. All figures in accepted manuscripts will be screened prior to publication.

Supplemental information: There are strict limits on the allowable amount of supplemental data. Articles may have up to 5 supplemental figures. Up to 10 supplemental videos or flash animations are allowed. A summary of all supplemental material should appear at the end of the Materials and methods section.

As you may know, the typical timeframe for revisions is three to four months. However, we at JCB realize that the implementation of social distancing and shelter in place measures that limit spread of COVID-19 also pose challenges to scientific researchers. Lab closures especially are preventing scientists from conducting experiments to further their research. Therefore, JCB has waived the revision time limit. We recommend that you reach out to the editors once your lab has reopened to decide on an appropriate time frame for resubmission. Please note that papers are generally considered through only one revision cycle, so any revised manuscript will likely be either accepted or rejected.

Thank you for this interesting contribution to Journal of Cell Biology. You can contact us at the journal office with any questions, cellbio@rockefeller.edu or call (212) 327-8588.

Sincerely,

Scott Emr, Ph.D.
Monitoring Editor

Marie Anne O'Donnell, Ph.D.
Scientific Editor

Journal of Cell Biology

Margaret S. Robinson
Emeritus Professor of Molecular Cell Biology

The Journal of Cell Biology
30 September, 2020

Dear Editors,

Thank you for the referees' comments on our manuscript, "Rag GTPases and Phosphatidylinositol 3-Phosphate Mediate Recruitment of the AP-5/SPG11/SPG15 Complex" (202002075). We were very happy to hear that our manuscript had received mainly positive reviews. It has taken us longer than we had anticipated to prepare a revised version. This is of course mainly due to the pandemic, and our lab being in complete lockdown from 20th March until 15th June. In addition, Jenny Hirst, who carried out all of the experimental work in the paper, was on full time secondment at the Cambridge COVID-19 testing facility. She continued to work there after our lab partially reopened, and only returned full-time at the end of August. Nevertheless, we have been able to generate new data for our manuscript, including results from our colleagues, Geoff Hesketh and Anne-Claude Gingras, who are now co-authors on the paper. We hope that this additional work will address the major concern of Reviewer 1 (see below).

From Reviewer 1:

The main concern is that all of the data employ cellular manipulations and assessment of SPG15, SPG11, or AP-5 by fluorescence/ immunofluorescence microscopy; there are no in vitro data or biochemical approaches to validate and extend the conclusions. In particular, the manuscript would benefit from biochemical evidence of an interaction - direct or not - between RagC-GDP and the SPG11/15 complex. Pull-downs in Suppl. Fig. 6B fail to detect binding of the AP-5 subunit AP5Z1, but neither SPG11 nor SPG15 were probed. It seems to me that this could easily be tested even if specific blotting antibodies for SPG11 or SPG15 were unavailable by exploiting the stable cell lines expressing GFP-SGP11 or -SPG15. One also wonders whether it might be possible to reconstitute the recruitment of tagged AP5/SPG11/SPG15 from cell lysates onto PtdIns3P-containing liposomes in the presence of recombinant RagC or RagC/RagA complex. Absent data such as this, the impact of knocking

down SPG11 but not AP-5 on SPG15 recruitment in Figure 6B-D is weak evidence that this complex mediates the RagC-GDP interaction, and all of the remaining data could be potentially explained by indirect interactions of RagC-associated factors with SPG11/15/AP-5. Even the requirement of the SPG15-FYVE domain interaction with PtdIns3P for AP-5 recruitment is not validated by data presented in the paper (see point 8 below).

We completely agree that biochemical data would strengthen our conclusion that SPG11/SPG15 interacts either directly or indirectly with GDP-bound RagC. The reviewer suggests that we try carrying out our Rag pulldowns (shown in our original Supplemental Fig. 6B but now deleted; see our response to point 1 from Reviewer 1) on cells expressing GFP-tagged constructs, and then probe blots with anti-GFP. We did in fact do this experiment, but did not see any specific labelling, and we now mention that in the text (page 14). Our inability to pull down AP-5/SPG11/SPG15 with Rags actually fits in well with what we know about other types of coats and the membrane-associated proteins that recruit them. These interactions tend to be both weak and conformation-dependent, causing the proteins to dissociate when cells are broken open. The use of detergents is also problematic, when the interactions depend at least in part on lipids. For instance, to the best of our knowledge, nobody has been able to co-immunoprecipitate APs or COPI with ARF1, even though the proteins interact on membranes and ARF1 is essential for the recruitment of all of these coats.

An approach that often works well for identifying transient interactions is proximity biotinylation, or BioID. This is a method carried out in living cells, in which a “bait” protein is fused to a promiscuous biotin ligase, BirA*. Upon addition of biotin to the medium, proteins that interact with and/or are in close proximity to the bait become biotinylated, and can be pulled down using streptavidin. We knew that a former colleague at the CIMR, Dr Geoffrey Hesketh, was carrying out large-scale proximity biotinylation experiments in the lab of Dr Anne-Claude Gingras, paying particular attention to players in the mTORC1 pathway. While the original version of our manuscript was in preparation, we contacted Geoff to discuss the implications of our findings. He showed us some of his own data, which fitted in remarkably well with the model we propose in Figure 7 – a model that we had already formulated before we contacted him.

Fortunately, Geoff Hesketh and Anne-Claude Gingras have allowed us to include some of their results in our manuscript, on which they are now co-authors. Our new Figure 5C shows Geoff’s proximity biotinylation data on cells expressing BirA*-tagged RagC: either wild-type, GDP-locked, or GTP-locked. His experiments show that only the GDP-locked form of RagC

biotinylates SPG15. A complete list of all the proteins biotinylated by the three forms of RagC is shown in Supplemental Table 1. We feel that these data provide strong biochemical support for our hypothesis, initially based on microscopy, that GDP-bound RagC facilitates the recruitment of the AP-5/SPG11/SPG15 complex.

Regarding whether or not the interaction with RagC is direct, we agree that this is a very important question, and one to which we do not yet have an answer. Therefore, we were very careful in the paper to make it clear that the Rag interaction could be mediated by another unidentified protein(s). We had already mentioned in the discussion that liposome binding experiments would probably be the best way to determine whether or not the interaction is direct, and we are planning to carry out these experiments in collaboration with our structural biology colleague, Dr Lauren Jackson at Vanderbilt University. We recently had a Zoom meeting with Lauren and her postdoc, and we will be sending them reagents, including constructs and cell lines, while they make the liposomes and carry out the binding assays. However, this will be a long-term project, with no guarantee of success, and we feel that it is beyond the scope of the present study.

The first referee also had some more minor comments:

1.) In general, many of the images shown are not particularly convincing of the changes in membrane recruitment described in the text and the quantification, particularly recruitment of SPG15/11/AP-5 in fed cells. There is substantial variation among the images in the level of recruitment and the number of puncta shown, particularly in Figures 4-6 and their Supplemental associated figures, and the images often do not match the quantification or the conclusions drawn in the text. A particularly egregious example is Suppl. Fig. 6A, which supposedly shows RagC-GDP-dependent stabilization of AP5Z1-GFP on membranes, but barely any puncta can be seen. The authors would benefit from ensuring that the most optimal images are shown, that the images are optimized (uniformly within a figure) for brightness/contrast, and that the images match the quantification. It would also benefit readers if the authors explicitly stated in the Results and/or Figure legends why some images show perinuclear accumulation whereas others show more peripheral accumulation (these fit the biology of the knockdowns).

We take the reviewer's point that there is variation amongst the images in the level of recruitment and number of puncta. The problem is that despite our best attempts, the cell lines are heterogeneous. All of the cells were sorted by flow cytometry and then clonal lines were isolated, and these clonal lines have been re-sorted, but there is still a lot of variability between

cells. This is why the quantification was so important, because it allowed us to objectively sample many cells in three separate experiments for each condition. So the quantitative results are really much more informative than the individual images. We now make a point about heterogeneity early on in the results (page 7). Regarding Supplemental Figure 6A, which shows AP5Z1-GFP cells, this cell line has always been the most problematic, and because these are fed cells, the discrete puncta can really only be seen in the RagC^{GDP}-expressing cells, where they colocalise with RagC^{GDP}. But because we were one supplemental figure over the limit, we have decided to delete Supplemental Figure 6 entirely. Regarding optimisation for brightness and contrast, we were very careful to process all images the same way (i.e., same exposure time and same manipulations), as we believe this is the only way they can be compared objectively, either by eye or electronically. This has now been clarified in the Materials and Methods (page 25). The reviewer also points out that some images show perinuclear accumulation while others show more peripheral accumulation, and that this is consistent with the biology of the knockdowns. That is absolutely right, and we didn't mention it because we were trying to keep our focus on membrane association. However, we now make this point in the legend for Figure 3A (page 35).

2.) All of the knockdowns employ only a single type of siRNA treatment (the Methods are a bit confusing, but they appear to be siRNA pools), and no rescue experiments are performed. The key experiments showing positive results (e.g. RagC knockdown) should be repeated with at least two separate individual siRNAs. Optimally rescue with an siRNA-resistant expression construct should also be done.

We agree that two ways of ensuring that knockdowns are specific are to use more than one siRNA against the same target, and/or include a rescue experiment. However, in this particular instance we obtained exactly the same phenotype when we knocked down Ragulator, which acts upstream from the Rags, and a similar (albeit weaker) phenotype when we knocked down RagA, which acts together with RagC. The fact that we got similar phenotypes when knocking down three components of the same pathway provides strong evidence that these effects are not off-target.

3.) For Figure 1 and Suppl. Fig. 1, the cell lines used and the origin of the GFP signal are not clear. Are these stably transduced HeLa cells in Fig. 1, S1A and S1C? It is indicated that the SPG15-GFP is expressed under its own promoter in some of these, but how? Are these knock-ins, or were the cells transfected with a construct containing a fragment of the SPG15 gene with GFP inserted in frame? These points need to be clarified in the text of the Results

(briefly), Materials and Methods (extensively, and the Figure legend (briefly). Additionally, Suppl. Fig. S1A and B would benefit from the addition of highly magnified insets to better demonstrate overlap (or not) with LAMP1 and EEA1.

We take the referee's point that the origins of these cell lines could have been more clearly explained. They were first published 10 years ago by Słabicki et al., who then generously shared them with us, and we say all this in the Materials and Methods. However, we agree that it would be helpful to say more about how the cells were actually made. They were generated using "BAC TransgeneOmics", a technique that was being developed at the time in Tony Hyman's and Frank Buchholz's labs in Dresden, both of whom were co-authors on the Słabicki et al. paper. We now go into more detail about the origin of the cells, in the Results (page 7), Materials and Methods (page 22), and legend to Figure 1. We also explain that in spite of cloning and re-sorting, the cells are heterogeneous (pages 7 and 22), and we include magnified inserts in Supplemental Figure 1.

4.) In Figure 2 and Suppl. Fig. 2, the following concerns should be addressed.

a.) In panel A, are all of the images taken at the same brightness/ contrast and similarly manipulated postacquisition? The signal for SPG15-GFP in cells treated with wortmannin seems much higher than the others.

Yes, as we explain above in the answer to Question 1, the images were always taken at the same brightness and contrast, but some of the cells express more SPG15-GFP than others, which is one of the reasons quantification was so important. The reason for the brighter signal in the cells treated with wortmannin is that most of the construct was cytosolic. Similarly, there is more of a cytosolic signal in fed cells than in starved cells or cells treated with a PI 5-kinase inhibitor, because there is less AP-5/SPG11/SPG15 on the membrane. We now make this clear in the figure legend (page 33).

b.) In panel B, it would be helpful if it were indicated on the Figure itself that the values correspond to SPG15- GFP and not to LAMP1.

This has been done.

c.) In the text describing the results, I suggest that the authors refrain from speculation on why the spots are bigger upon treatment with the PIKfyve inhibitor until the Discussion. While the explanation posited is plausible, given that PtdIns(3,5)P2 activates TRPML and TPC channels

on lysosomes that are required for lysosomal metabolic maintenance, it is equally (or perhaps more) likely that PIKfyve inhibition results in indirect inhibition of mTORC1 which then leads to increased AP-5 recruitment.

We thank the referee for the suggestion, and have removed the speculation from the Results section (page 8). In the figure legend, we now mention that some of the effects of PI5K inhibition may be indirect (page 34).

d.) In Suppl. Fig. 2B, the PI5K inhibitor treatments need to be compared to controls without the inhibitor.

The figure has now been expanded to include controls without the inhibitor.

e.) The Results at bottom of page 8 cite Figure 2C as indicative that GFP-FYVE(SPG15) is expressed at near endogenous levels, but no evidence (at least here, and I could not find it in the cited references) is provided that the full-length GFP-SPG15 is expressed at endogenous levels. Either provide the proof or remove the statement.

The reason we said that GFP-FYVE(SPG15) is expressed at near-endogenous levels is that Figure 2C shows that it is expressed at similar levels to the full-length GFP-SPG15, and the full-length construct makes use of its endogenous promoter and was shown to be expressed at near-endogenous levels in the first paper that described this construct, Słabicki et al. 2010. We now simply say that the GFP-FYVE(SPG15) construct was expressed at similar levels to the full-length construct (page 8).

f.) Panels in Figure 2E would benefit from highly magnified insets to emphasize the colocalization or lack thereof.

This has been done.

5.) In Figure 3F, how many cells/ profiles in how many experiments were quantified?

As we explain in the Materials and Methods (page 25), the cells were always quantified in the same way: a minimum of 20 cells were analysed for each condition and repeated at least three times. We now added that information to the legend for Figure 3F.

6.) *Figure 4 and Suppl. Fig. S4 would benefit from quantification of # of puncta/ cell as in previous figures. Alternatively, membrane association of AP-5 in the cell population could be assessed biochemically by cell fractionation to membrane and cytosol fractions.*

We did try to quantify the number of puncta per cell for the NPRL3 and FLCN knockdowns, even though no difference was discernable by eye, but the data were inconclusive. In the end, we decided to quantify only those conditions where there was a robust difference. We also tested – for several of our conditions – whether changes in localisation could be correlated with changes in the soluble pool, as determined by homogenisation and centrifugation, exactly as suggested by the reviewer. However, even in wortmannin-treated cells, there was no clear difference in the amount of AP-5/SPG11/SPG15 in the soluble fraction. This may have to do with the behaviour of the complex in broken-open cells. A parallel situation arose nearly 30 years ago, when we tried and failed to find differences in the soluble fraction of AP-1 and COPI in brefeldin A-treated cells (see Robinson and Kreis, 1992, PMID 1555237), even though the drug caused both types of coats to be lost from membranes.

7.) *In Suppl. Fig. S5A, is the lower left panel of the fed cells mTOR in the same cells as those shown to the right?*

Yes, we have now added the missing label and thank the referee for pointing this out

8.) *On page 14 in the Results, the authors state: "The PI3P interaction [of the AP-5/SPG11/15 complex] is clearly mediated by the SPG15 FYVE domain", but this is not supported by the data shown - all that is shown is that localization of SPG15 itself is ablated by a mutation predicted to interfere with FYVE binding to PtdIns3P. Indeed, it would be a good experiment to express either the intact or R183A mutant form of HA-SPG15 in cells knocked down for endogenous SPG15 and expressing SPG11-GFP or tagged AP-5 subunits to test whether this statement is true. In the absence of such data, the conclusion cannot be drawn.*

Although we have not tried to localise the other subunits in cells expressing the R183A mutant form of SPG15, we do have evidence from earlier studies that wortmannin causes the loss of tagged μ 5 and σ 5 from the membrane. We have now toned down the sentence to say, "The PI3P interaction is a function of the SPG15 FYVE domain" (page 14).

9.) *The observation that overexpression of RagC-GDP but not of RagD-GDP induces SPG15 recruitment is interesting; would this not provide a nice potential approach to identify a*

potential interacting region of RagC for the complex through domain swaps between RagC and RagD?

This is something we are pursuing, and in fact these experiments were underway when our lab was shut down. Our first concern was that there might be a more trivial explanation for the difference between RagC^{GDP} and RagD^{GDP}, because the two constructs (RagC75L and RagD77L), which were kindly provided by the Sabatini lab, were not entirely equivalent. There is a highly conserved stretch of residues near the N terminus containing the P loop, which plays a key role in nucleotide binding, and which is identical in RagC and RagD: GLRRSGKSSI. Both Rags had serine-to-leucine substitutions, but the sequence was GLRRSGKLSI in RagC75L and GLRRSGKSLI in RagD77L. One of our colleagues, who had carried out structural studies on the Rags, told us that the first serine was more important for nucleotide binding, so we made the GLRRSGKLSI mutation in RagD (RagD76L). We obtained similar results, with neither the RagD76L nor the RagD77L mutant promoting SPG15-GFP recruitment in fed cells, although both constructs promoted the recruitment of mTOR.

Since returning to the lab, Jenny has started to make chimeras, focusing on the unstructured N and C termini of Rags C and D, as they are the most divergent (see below). So far, she has made three constructs, all with the central portion derived from RagD: CDD (i.e., N-terminal RagC, central RagD, and C-terminal RagD), DDC, and CDC. All of these constructs were expressed at comparable levels, but none of the chimeras was able to promote recruitment of SPG15-GFP in fed cells. This indicates that the conserved central portion of RagC contributes to recruitment, possibly together with the N and/or C termini. We intend to carry out further

dissections, but because there doesn't appear to be a simple answer to the question of why RagC promotes recruitment while RagD does not, we feel that these experiments are beyond the scope of our paper.

RagC	MSLQYGAEEETPLAGSYGAADSFPKDFGYVVEEEEEAAAAAGGGVAGAGGGCGP	SS-----	60			
RagD	MSQVLGKPKQ-----PQEDDAE	EEEEETEL-----VGL-ADYGDGPDSSADPDSGTEEGVLD	FSPFSTE	61		
RagC	RFILLMGLRS	SGKSSIQVVFHMSPN	ETLESTNKIYKDDISNSSFVNFQI	WDFPGQMEFFDPTFDYEMIFRGT	GAL	140
RagD	RFILLMGLRS	SGKSSIQVVFHMSPN	ETLESTNKICEDVSNSSFVNFQI	WDFPGQIDFFDPTFDYEMIFRGT	GAL	141
RagC	IYVILAQDDYMEALTRLHITVSKAYKVN	PDMMNFVFIHKVVDGLSDDHRIETQ	DIHQRANDDLADAGLEKRLS	FYLTSI	220	
RagD	IFVILSODDYMEALTRLHITVTRAYKVN	TDINFEVFIHKVVDGLSDDHRIETQ	DIHQRANDDLADAGLEKRLS	FYLTSI	221	
RagC	YDHSIFEAFSKV	VQKLIPLQLTLENLLNIFISNSGIE	NAFLFVVSIIYIATDSSPV	MQSYELCCMI	VVIVVSCIY	300
RagD	YDHSIFEAFSKV	VQKLIPLQLTLENLLNIFISNSGIE	NAFLFVVSIIYIATDSTPV	MQTYELCCMI	VVIVVSCIY	301
RagC	LKEDGSGSAYDKESMAII	LNNTTVLYLKEVTKFLALVCILREES	FERKGLIDYNFHCFRKAI	HEVFEVGVTS	HSKSCGHQ	380
RagD	LKEDGAGTPYDKESTAI	LNNTTVLYLKEVTKFLALVCVREES	FERKGLIDYNFHCFRKAI	HEVFEVRMKVV	SRKVC	381
RagC	TSASSL	ALTHNGTP	NAI	399		
RagD	NRLQKK	RATPN	GTPLVLL	400		

10.) *The authors provide a rather unsatisfying discussion of the paradoxical recruitment of SPG15 to lysosomes by starvation but also by overexpression of RagC-GDP - confusing given that starvation is thought to induce GTP exchange onto RagC. Might this imply that the SPG11/15/AP-5 complex is recruited at a time prior to full activation of the Rags (e.g. by GEF-mediated activation of RagA/B) but after partial activation through GTPase activity on RagC?*

The referee makes an important point here. Although it is tempting to speculate that the molecular mechanism for increased recruitment of SPG15 onto membranes is always the same, the similar effects of starvation and overexpression of RagC^{GDP} are indeed paradoxical. We were careful not to imply that the mechanisms were necessarily the same, and our Discussion we included the sentence “Precisely how this apparent preference of the AP-5/SPG11/SPG15 complex for both Rags in their GDP-bound state fits in with its increased recruitment during starvation is still unclear.” The referee’s suggestion is one that we had considered and we now include it on page 17.

11.) *More details are needed in the Materials and Methods for the quantification of particle size and number in Figures 2, 3, 5, 6 and S5.*

We now include these details in the Materials and Methods section (page 25).

Reviewer 2:

1. In Figure 3A the Arl18 knockdown does look like it is causing a redistribution of SPG15-GFP.

Reviewer 1 also commented on how some knockdowns cause increased perinuclear labelling while others cause increased peripheral labelling, and Arl8 is a good example of a knockdown that causes an increase in perinuclear labelling. Although we wanted to keep our focus on how much of the SPG15-GFP was membrane-associated, rather than where those membranes were, we now acknowledge that the Arl8 knockdown causes a redistribution of SPG15-GFP in the figure legend (page 35).

2. The interpretation of Figure 6C is not clear from the images shown but seems clear in the analysis of many cells. If the analysis is correct, maybe different fields should be shown in 6C.

Although there is still some weak punctate labeling in the SPG11 knockdown cells in Figure 6C, this is consistent with the graph in Figure 6D, which shows that the spot number is reduced but does not go down to zero. What is clearer from the immunofluorescence image is that the relative amount of cytosolic labelling is increased, and we now make this point in the figure legend.

Reviewer 3:

1. The major deficit of this manuscript is the complete absence of functional data. The authors have previously assigned a role for the AP-5/SPG11/SPG15 complex in protein sorting, and now they speculate that it could be involved in nutrient signaling. However, in the absence of data that show the involvement of AP-5/SPG11/SPG15 in nutrient signaling, that this depends on PI3P/Rag coincidence detection, and that it has consequences for the cell, the impact of the present findings is moderate.

We agree that it will be important to investigate the potential role of the AP-5/SPG11/SPG15 complex in nutrient signalling. We have been monitoring mTOR signalling under various conditions in embryonic fibroblasts from control, SPG11 knockout, SPG15 knockout, and AP5Z1 knockout mice, using phosphorylation of S6K as a readout. Initially there appeared to be a difference in the cells' ability to reactivate mTOR after prolonged starvation. However, there were also differences in cell growth, and further studies in which the growth conditions were normalised as much as possible did not show a consistent effect. Thus, unravelling the

interplay between AP-5/SPG11/SPG15 and nutrient signalling will not be straightforward, and we feel that it is beyond the scope of this paper.

2. The authors have previously shown that wortmannin prevents recruitment of the AP-5/SPG11/SPG15 complex to endosomes (Hirst et al., MBC, 2013), so the aspect of PI3P dependence is not novel. Additionally it is not clear why the authors did not choose to use more subclass specific PI3K inhibitors than wortmannin.

We realise that the wortmannin result is not novel, but we have followed it up by making mutations in the SPG15 FYVE domain that prevent PI3P binding, which we feel is a more rigorous test of PI3P dependence than investigating other inhibitors. Moreover, our finding that the SPG15 FYVE domain on its own goes to early endosomes rather than late endosomes is completely novel, and an important extension of our previous findings.

3. The coincidence detection of PI3P and Rags by SPG15 is novel and interesting, but further molecular insight is required. Whereas the FYVE domain of SPG15 clearly recognizes PI3P, the authors should use deletion mutagenesis to establish which domain binds Rags.

This is something we intend to do, but SPG11 and SPG15 are both very large proteins of ~280 kD, and the only clear domains are the FYVE domain in SPG15 and the putative β -propeller in SPG11. Otherwise, the two proteins are almost entirely α -solenoid. Based on other large α -solenoid-containing proteins involved in membrane traffic (e.g., clathrin heavy chain, COPI α and β' subunits, and Sec31), we suspect that the α -solenoids of SPG11 and SPG15 interact extensively with each to form an obligatory heterodimer. Thus, deletion mutants are likely to be non-functional. Nevertheless, we did try to make several truncations of SPG15; however, all of these constructs were cytosolic. Probably the only way to establish how the SPG11/SPG15 complex binds to Rags would be to express the entire complex and carry out structural studies. This is something we plan to do in collaboration with Lauren Jackson's lab, but it will be a major undertaking that is likely to take years rather than months.

(Minor points)

1. It is difficult to understand how SPG15 could be recruited to endosomes/lysosomes by coincidence detection of PI3P and Rags as long as the FYVE domain alone of SPG15 is sufficient to localize to endosomes. Do the authors think that there is allosteric regulation of AP-5/SPG11/SPG15 interaction with endosomes?

The reviewer makes an interesting and important point here: the PI3P interaction is sufficient to target the SPG15 FYVE domain on its own to membranes, but not in the context of the whole complex. There are precedents for coincidence detection involving allosteric regulation; probably the best characterised of these is the conformational change that occurs in AP-2 when it comes in contact with the plasma membrane, causing binding sites for PIP2, cargo, and clathrin to be exposed. We now discuss this in the text (page 20).

2. "Starvation" could mean many things, and I could not find how it was actually done. This should be mentioned explicitly in the Results, with details in Figure legend or MM.

We now specify how we did the starvation in the Results section (page 8) and the first figure legend (we actually did specify how we did it in the Materials and Methods, on page 23).

Finally, we thank the reviewers for their constructive suggestions, which we feel have improved our manuscript. We hope that with the changes we have made, the referees will recommend publication. Thank you again for considering our manuscript, and we look forward to hearing from you

Yours sincerely,

Margaret S. Robinson

Jennifer Hirst

CIMR, Cambridge CB2 0XY, UK
Email: msr12@cam.ac.uk, jh228@cam.ac.uk

October 23, 2020

RE: JCB Manuscript #202002075RR

Prof. Margaret S Robinson
University of Cambridge
CIMR
Cambridge CB2 0XY
United Kingdom

Dear Scottie:

Thank you for submitting your revised manuscript entitled "Rag GTPases and Phosphatidylinositol 3-Phosphate Mediate Recruitment of the AP-5/SPG11/SPG15 Complex". The paper has been assessed again by the original reviewers. As you will see, reviewers #1 and #2 are largely satisfied with the revisions and now recommend acceptance. However, reviewer #3 continues to feel that the study does not represent a sufficient advance to warrant publication in JCB. While we appreciate this reviewer's position and recognize why s/he feels this way, we disagree with this assessment. Therefore, we would be happy to publish your paper in JCB pending final revisions necessary to meet our formatting guidelines (see details below).

*As you will also see, reviewer #1 has raised several relatively minor concerns which we would like for you to address in the final revision. Addressing these issues should not require any new experiments. However, please be sure to include a point-by-point rebuttal to these final concerns along with your revised manuscript.**

A. MANUSCRIPT ORGANIZATION AND FORMATTING:

Full guidelines are available on our Instructions for Authors page, <https://jcb.rupress.org/submission-guidelines#revised>. **Submission of a paper that does not conform to JCB guidelines will delay the acceptance of your manuscript.**

1) Text limits: Character count for Articles and Tools is < 40,000, not including spaces. Count includes title page, abstract, introduction, results, discussion, and acknowledgments. Count does not include materials and methods, figure legends, references, tables, or supplemental legends. You are currently below this limit but please bear it in mind when revising.

2) Figure formatting: Scale bars must be present on all microscopy images, including inset magnifications. Molecular weight or nucleic acid size markers must be included on all gel electrophoresis.

3) Statistical analysis: Error bars on graphic representations of numerical data must be clearly described in the figure legend. The number of independent data points (n) represented in a graph must be indicated in the legend. Statistical methods should be explained in full in the materials and

methods. For figures presenting pooled data the statistical measure should be defined in the figure legends. Please also be sure to indicate the statistical tests used in each of your experiments (both in the figure legend itself and in a separate methods section) as well as the parameters of the test (for example, if you ran a t-test, please indicate if it was one- or two-sided, etc.). Also, since you used parametric tests in your study (e.g. t-tests, ANOVA, etc.), you should have first determined whether the data was normally distributed before selecting that test. In the stats section of the methods, please indicate how you tested for normality. If you did not test for normality, you must state something to the effect that "Data distribution was assumed to be normal but this was not formally tested."

4) Materials and methods: Should be comprehensive and not simply reference a previous publication for details on how an experiment was performed. Please provide full descriptions (at least in brief) in the text for readers who may not have access to referenced manuscripts. The text should not refer to methods "...as previously described."

5) Please be sure to provide the sequences for all of your primers/oligos and RNAi constructs in the materials and methods. You must also indicate in the methods the source, species, and catalog numbers (where appropriate) for all of your antibodies.

6) Microscope image acquisition: The following information must be provided about the acquisition and processing of images:

- a. Make and model of microscope
- b. Type, magnification, and numerical aperture of the objective lenses
- c. Temperature
- d. imaging medium
- e. Fluorochromes
- f. Camera make and model
- g. Acquisition software
- h. Any software used for image processing subsequent to data acquisition. Please include details and types of operations involved (e.g., type of deconvolution, 3D reconstitutions, surface or volume rendering, gamma adjustments, etc.).

7) References: There is no limit to the number of references cited in a manuscript. References should be cited parenthetically in the text by author and year of publication. Abbreviate the names of journals according to PubMed.

8) Supplemental materials: There are strict limits on the allowable amount of supplemental data. Articles/Tools may have up to 5 supplemental figures. At the moment, you meet this limit but please bear it in mind when revising.

Please also note that tables, like figures, should be provided as individual, editable files. A summary of all supplemental material should appear at the end of the Materials and methods section.

9) Conflict of interest statement: JCB requires inclusion of a statement in the acknowledgements regarding competing financial interests. If no competing financial interests exist, please include the following statement: "The authors declare no competing financial interests." If competing interests are declared, please follow your statement of these competing interests with the following statement: "The authors declare no further competing financial interests."

10) A separate author contribution section is required following the Acknowledgments in all research manuscripts. All authors should be mentioned and designated by their first and middle

initials and full surnames. We encourage use of the CRediT nomenclature (<https://casrai.org/credit/>).

11) ORCID IDs: ORCID IDs are unique identifiers allowing researchers to create a record of their various scholarly contributions in a single place. At resubmission of your final files, please consider providing an ORCID ID for as many contributing authors as possible.

B. FINAL FILES:

-- High-resolution figure and video files: See our detailed guidelines for preparing your production-ready images, <https://jcb.rupress.org/fig-vid-guidelines>.

****It is JCB policy that if requested, original data images must be made available to the editors. Failure to provide original images upon request will result in unavoidable delays in publication. Please ensure that you have access to all original data images prior to final submission.****

****The license to publish form must be signed before your manuscript can be sent to production. A link to the electronic license to publish form will be sent to the corresponding author only. Please take a moment to check your funder requirements before choosing the appropriate license.****

Thank you for this interesting contribution, we look forward to publishing your paper in Journal of Cell Biology.

Sincerely,

Scott Emr, PhD
Monitoring Editor
Journal of Cell Biology

Tim Spencer, PhD

Reviewer #1 (Comments to the Authors (Required)):

In their revised manuscript and the accompanying response to reviewers, Hirst et al have satisfactorily addressed most of the issues raised in the initial review. Some new concerns, described below (point 1), arise from the presentation of the new added data in Figure 5 and Suppl. Table 1. These concerns can be addressed by alterations to the text. In addition, there are two minor corrections to the text that need to be made (points 2 and 3). With these changes, the manuscript will make an important contribution to the literature and to the JCB.

1. The proximity biotinylation experiment is a good addition to the manuscript. However, there are several concerns with the way that the data are presented and interpreted. First, in the Suppl. Table 1 SPG15 is listed by its official gene name, ZYVE26, which is not mentioned anywhere in the Figure 5 legend, Suppl. Table 1 legend, or the main text. Unless they have similarly cryptic names, none of the other SPG15/AP-5 complex components are present in this database; this should be noted and discussed. Second, FDR is not defined anywhere, nor is it explained why a value <1 is essential for a meaningful result. Third, I could not for the life of me understand the order in which the hits were listed in Suppl. Table 1; personally, I could not make sense of it until I sorted on the basis of the Fold Change (column L) or logOddsScore (column K). Fourth, having done that, the fold change/ logOddsScore of SPG15 peptide recovery from labeling by the GDP-locked RagC form relative to the GTP locked mutant is quite low compared to other RagC hits (e.g. mTORC1 and Ragulator/LAMTOR subunits); this again should be noted and discussed. Finally, and most critically, examination of the Table shows that many of the hits with similar scores are known lysosomal proteins that indeed localize to the same organelle as RagC, but few of which are likely interactors of RagC or the Ragulator/ mTORC1 super-complex. This raises concerns that the bioID experiment simply identifies global late endosomal/ lysosomal membrane-associated contents rather than components that specifically interact with the Rags. Thus, while this experiment provides a nice non-microscopy-based approach to confirm the localization of SPG15 to late endosomes/ lysosomes, conclusions regarding the proximity of the SPG15/11/AP-5 complex to RagC should be tempered, and some discussion should be added regarding why other members of the SPG15/11/AP-5 complex were not identified in the bioID experiment (if that is true) and whether the other proteins identified represent either a relevant subset or a global survey of lysosomal membrane-associated proteins.

2. References in the text (page 13) to panel E of Figure 5 need to be revised given the addition of panel C.

3. Page 14 still includes a reference to Supplemental Figure 6A, which has been removed.

Reviewer #2 (Comments to the Authors (Required)):

No further comments.

Reviewer #3 (Comments to the Authors (Required)):

I maintain my view that, in the absence of functional data, this manuscript does not represent a sufficient advance for publication in JCB, although the technical quality is good. Neither of the major points I raised was fully addressed, either because experiments did not work or because the authors considered the point beyond the scope of the manuscript.

Margaret S. Robinson
Emeritus Professor of Molecular Cell Biology

The Journal of Cell Biology
23 November, 2020

Dear Editors,

Thank you for the comments on our manuscript, "Rag GTPases and Phosphatidylinositol 3-Phosphate Mediate Recruitment of the AP-5/SPG11/SPG15 Complex" (202002075RR). We were very happy to hear that our manuscript is now acceptable for publication in *JCB*. As you requested, we have addressed the minor concerns raised by Reviewer 1, as described below.

The proximity biotinylation experiment is a good addition to the manuscript. However, there are several concerns with the way that the data are presented and interpreted. First, in the Suppl. Table 1 SPG15 is listed by its official gene name, ZFYVE26, which is not mentioned anywhere in the Figure 5 legend, Suppl. Table 1 legend, or the main text.

The name ZFYVE26 is now mentioned in the main text (page 5), in the legend to Figure 5, and in the Supplemental Table 1 legend.

Unless they have similarly cryptic names, none of the other SPG15/AP-5 complex components are present in this database; this should be noted and discussed.

The referee makes an interesting point, and we now discuss it on pages 12-13. The low abundance of all of the components of the complex is clearly a factor: e.g., in the HeLa spatial proteome database, <http://mapofthecell.biochem.mpg.de/>, only SPG15/ZFYVE26, SPG11, and AP5Z1 were detected, and only in one or two of the five maps. However, the lack of other components in the complex in proximity biotinylation database may also be a result of their proximity to RagC relative to that of SPG15, something we hope to address in the future.

Second, FDR is not defined anywhere, nor is it explained why a value <1 is essential for a meaningful result.

FDR (false discovery rate) is now defined in the Methods section. 1% FDR has been historically used in the Gingras lab as a cut-off for 'high confidence' interactions and was arrived at empirically through analysis of multiple proteomic datasets. Typically, this 1% FDR cut-off is used when a threshold is required to select a subset of 'high confidence' interactors from a larger dataset for further bioinformatics analysis (for example, to look for functional enrichment of specific GO terms, pathways, protein domains, etc.). However, in the current paper we do not do any thresholding (or further bioinformatic analysis) of the dataset, and report the BioID results to show that the biotinylation profiles are consistent with the cell biology data reported throughout the paper. Therefore, the notion of 'high confidence' interactions does not apply to the BioID data as presented in this paper and we have therefore removed this language from the main text.

Third, I could not for the life of me understand the order in which the hits were listed in Supp. Table 1; personally, I could not make sense of it until I sorted on the basis of the Fold Change (column L) or logOddsScore (column K).

The purpose of Table 1 is to report the results of the complete BioID dataset, and to provide the values that underlie the selected interactions reported in the Fig 5C dotplot (highlighted in red in the table). Readers are free to order / analyze the data in the Excel table in whatever way best suits their needs. We have bolded preys with an FDR of 1% or less simply to highlight those preys that are most significantly enriched above background. As a starting point, we have now ordered the bolded preys based on average spectral counts (high to low), and this is indicated in the Table legend.

Fourth, having done that, the fold change/ logOddsScore of SPG15 peptide recovery from labeling by the GDP-locked RagC form relative to the GTP locked mutant is quite low compared to other RagC hits (e.g. mTORC1 and Ragulator/LAMTOR subunits); this again should be noted and discussed.

This is likely due to the low abundance of SPG15 relative to RPTOR and LAMTOR1 in cells, and is discussed in the main text (page 12).

Finally, and most critically, examination of the Table shows that many of the hits with similar scores are known lysosomal proteins that indeed localize to the same organelle as RagC, but few of which are likely interactors of RagC or the Ragulator/ mTORC1 super-complex. This

raises concerns that the bioID experiment simply identifies global late endosomal/ lysosomal membrane-associated contents rather than components that specifically interact with the Rags. Thus, while this experiment provides a nice non-microscopy-based approach to confirm the localization of SPG15 to late endosomes/ lysosomes, conclusions regarding the proximity of the SPG15/11/AP-5 complex to RagC should be tempered, and some discussion should be added regarding why other members of the SPG15/11/AP-5 complex were not identified in the bioID experiment (if that is true) and whether the other proteins identified represent either a relevant subset or a global survey of lysosomal membrane-associated proteins.

The Reviewer is correct to point out that BioID has the potential to capture prey proteins that reside within the same membrane compartment as the bait protein, and may not necessarily represent direct interactions. Nonetheless, BioID does provide a measure of the distance relationship between a prey and bait — the closer a prey is to a bait the greater the extent to which it can be biotinylated. Importantly, all three forms of RagC (WT, GDP-locked, GTP-locked) were able to biotinylate LAMTOR1 to similar degrees. LAMTOR1 is the membrane-anchored subunit of the Ragulator complex that docks the Rag heterodimer on the lysosome surface. Therefore, the BioID data are consistent with all three RagC baits localizing as expected to the lysosome surface. Instead, only a small handful of proteins were labelled equally well by all three forms (we have used BioID to show that the global lysosomal proteome is several hundreds of proteins; see Hesketh et al., 2020, PMID 3306361). We therefore do suspect that the proteins labelled by specific nucleotide-binding states of RagC represent relevant subsets. Furthermore, the SPG15 biotinylation profile mirrors that of another protein recruited to the lysosome surface through direct binding to RagC-GDP (i.e., Raptor), although we do acknowledge that our BioID assay cannot establish whether SPG15 binds directly to RagC-GDP, or whether recruitment may be indirect. Nonetheless, our data support the notion that SPG15 comes into close proximity (whether direct or indirect) to RagC-GDP, but not to RagC-WT. We have been careful in the text not to imply that our BioID data is evidence of a direct interaction.

2. References in the text (page 13) to panel E of Figure 5 need to be revised given the addition of panel C.

We thank the referee for spotting this, and we have now made the necessary change.

3. Page 14 still includes a reference to Supplemental Figure 6A, which has been removed.

Again, we thank the referee for spotting this, and we have removed the reference to Supplemental Figure 6A.

We hope that we now have done everything we need for final approval of the manuscript.

Yours sincerely,

Margaret S. Robinson

Jennifer Hirst

CIMR, Cambridge CB2 0XY, UK
Email: msr12@cam.ac.uk, jh228@cam.ac.uk